# Model-Based Opponent Modeling

Xiaopeng Yu    Jiechuan Jiang    Wanpeng Zhang    Haobin Jiang    Zongqing Lu[†]

School of Computer Science, Peking University

## Abstract

When one agent interacts with a multi-agent environment, it is challenging to deal with various opponents unseen before. Modeling the behaviors, goals, or beliefs of opponents could help the agent adjust its policy to adapt to different opponents. In addition, it is also important to consider opponents who are learning simultaneously or capable of reasoning. However, existing work usually tackles only one of the aforementioned types of opponents. In this paper, we propose *model-based opponent modeling* (MBOM), which employs the environment model to adapt to all kinds of opponents. MBOM simulates the recursive reasoning process in the environment model and imagines a set of improving opponent policies. To effectively and accurately represent the opponent policy, MBOM further mixes the imagined opponent policies according to the similarity with the real behaviors of opponents. Empirically, we show that MBOM achieves more effective adaptation than existing methods in a variety of tasks, respectively with different types of opponents, *i.e.*, fixed policy, naïve learner, and reasoning learner.

## 1   Introduction

Reinforcement learning (RL) has made great progress in multi-agent competitive games, *e.g.*, AlphaGo [33], OpenAI Five [26], and AlphaStar [39]. In multi-agent environments, an agent usually has to compete against or cooperate with diverse other agents (collectively termed as *opponents* whether collaborators or competitors) unseen before. Since the opponent policy influences the transition dynamics experienced by the agent, interacting with diverse opponents makes the environment nonstationary from the agent's perspective. Due to the complexity and diversity in opponent policies, it is very challenging for the agent to retain overall supremacy.

Explicitly modeling the behaviors, goals, or beliefs of opponents [2], rather than treating them as a part of the environment, could help the agent adjust its policy to adapt to different opponents. Many studies rely on predicting the actions [11, 13, 10, 27] and goals [29, 30] of opponents during training. When facing diverse or unseen opponents, the agent policy conditions on such predictions or representations generated by corresponding modules. However, opponents may also have the same reasoning ability, *e.g.*, an opponent who makes predictions about the agent's goal. In this scenario, higher-level reasoning and some other modeling techniques are required to handle such sophisticated opponents [40, 43, 45]. In addition, the opponents may learn simultaneously, the modeling becomes unstable, and the fitted models with historical experiences lag behind. To enable the agent to continuously adapt to learning opponents, LOLA [8] takes into account the gradients of the opponent's learning for policy updates, Meta-PG [1] formulates continuous adaptation as a meta-learning problem, and Meta-MAPG [16] combines meta-learning with LOLA. However, LOLA requires knowing the learning algorithm of opponents, while Meta-PG and Meta-MAPG require all opponents use the same learning algorithm.

---

[†]Correspondence to ✉ zongqing.lu@pku.edu.cn

36th Conference on Neural Information Processing Systems (NeurIPS 2022).

Unlike existing work, we do not make such assumptions and focus on enabling the agent to learn effectively by directly representing the policy improvement of opponents when interacting with them, even if they may be also capable of reasoning. Inspired by the intuition that humans could anticipate the future behaviors of opponents by simulating the interactions in the brain after knowing the rules and mechanics of the environment, in this paper, we propose ***model-based opponent modeling*** (**MBOM**), which employs the environment model to predict and capture the policy improvement of opponents. By simulating the interactions in the environment model, we could obtain the best responses of opponents to the agent policy that is conditioned on the opponent model. Then, the opponent model can be finetuned using the simulated best responses to get a higher-level opponent model. By recursively repeating the simulation and finetuning, MBOM imagines the learning and reasoning of opponents and generates a set of opponent models with different levels, which could also be seen as recursive reasoning. However, since the learning and reasoning of opponents are *unknown*, a certain-level opponent model might erroneously estimate the opponent. To effectively and accurately represent the opponent policy, we further propose to mix the imagined opponent policies according to the similarity with the real behaviors of opponents updated by the Bayesian rule.

We empirically evaluate MBOM in a variety of competitive tasks, against three types of opponents, *i.e.*, fixed policy, naïve learner, and reasoning learner. MBOM outperforms strong baselines, especially when against naïve learner and reasoning learner. Ablation studies verify the effectiveness of recursive imagination and Bayesian mixing. We also show that MBOM can be applied in cooperative tasks.

## 2 Related Work

**Opponent Modeling.** In multi-agent reinforcement learning (MARL), it is a big challenge to form robust policies due to the unknown opponent policy. From the perspective of an agent, if opponents are considered as a part of the environment, the environment is unstable and complex for policy learning when the policies of opponents are also changing. If the information about opponents is included, *e.g.*, behaviors, goals, and beliefs, the environment may become stable, and the agent could learn using single-agent RL methods. This line of research is *opponent modeling*.

One simple idea of opponent modeling is to build a model each time a new opponent or group of opponents is encountered [48]. However, learning a model every time is not efficient. A more computationally tractable approach is to represent an opponent's policy with an embedding vector. Grover et al. [10] uses a neural network as an encoder, taking as input the trajectory of one agent. Imitation learning and contrastive learning are used to train the encoder. Then, the learned encoder can be combined with RL by feeding the generated representation into policy or/and value networks. Learning of the model can also be performed simultaneously with RL, as an auxiliary task [14]. Based on DQN [24], DRON [11] and DPIQN [13] use a secondary network that takes observations as input and predicts opponents' actions. The hidden layer of this network is used by the DQN module to condition on for a better policy. It is also feasible to model opponents using variational auto-encoders [27], which means the generated representations are no longer deterministic vectors, but high-dimensional distributions. ToMnet [29] tries to make agents have the same Theory of Mind [28] as humans. ToMnet consists of three networks, reasoning about the agent's action and goal based on past and current information. SOM [30] implements Theory of Mind from a different perspective. SOM uses its own policy, opponent's observation, and opponent's action to work backward to learn opponent's goal by gradient ascent.

The methods aforementioned only consider opponent policies that are independent of the agent. If opponents hold a belief about the agent, the agent can form a higher-level belief about opponents' beliefs. This process can perform recursively, termed *recursive reasoning*. In repeated games, R2-B2 [5] performs recursive reasoning by assuming that all agents select actions according to GP-UCB acquisition function [34], and shows that recursive reasoning on one level higher than the other agents leads to faster regret convergence. When it comes to more complicated stochastic games, recursive reasoning should be compatible with reinforcement learning algorithms. PR2 [40] and GR2 [41] use the agent's joint Q-function to obtain recursive reasoning. Yuan et al. [45] takes both level-0 and level-1 beliefs as input to the value function, where level-0 belief is updated according to the Bayesian rule and level-1 belief is updated using a learnable neural network. However, these methods [40, 41, 45] use centralized training with decentralized execution algorithms to train a set of fixed agents that cannot handle diverse opponents in execution.

If the opponents are also learning, the modeling mentioned above becomes unstable and the fitted models with historical experiences lag behind. So it is beneficial to take into consideration the learning process of opponents. LOLA [8] introduces the impact of the agent's policy on the anticipated parameter update of the opponent. A neural network is used to model the opponent's policy and estimate the learning gradient of the opponent's policy, implying that the learning algorithm used by the opponent should be known, otherwise the estimated gradient will be inaccurate. Further, the opponents may still be learning continuously during execution. Meta-PG [1] is a method based on meta policy gradient, using trajectories from the current opponents to do multiple meta-gradient steps and construct a policy that is good for the updated opponents. Meta-MAPG [16] extends this method by including an additional term that accounts for the impact of the agent's current policy on the future policies of opponents, similar to LOLA. These meta-learning based methods require that the distribution of trajectories matches across training and test, which implicitly means all opponents use the same learning algorithm.

Unlike existing work, we go one step further and consider a more general setting, where the opponents could be fixed, randomly sampled from an unknowable policy set, or continuously learning using an unknowable and changeable RL algorithm, both in training and execution. We learn an environment model that allows the agent to perform recursive reasoning against opponents who may also have the same reasoning ability.

**Model-Based RL and MARL.** Model-based RL allows the agent to have access to the transition function. There are two typical branches of model-based RL approaches: background planning and decision-time planning. In background planning, the agent could use the learned model to generate additional experiences for assisting learning. For example, Dyna-style algorithms [35, 19, 23] perform policy optimization on simulated experiences, and model-augmented value expansion algorithms [7, 25, 3] use model-based rollouts to improve the update targets. In decision-time planning [4], the agent could use the model to rollout the optimal action at a given state by looking forward during execution, *e.g.*, model predictive control. Recent studies have extended model-based methods to multi-agent settings for sample efficiency [46, 47], centralized training [42], and communication [17]. Unlike these studies, we exploit the environment model for opponent modeling.

## 3 Preliminaries

In general, we consider an $n$-agent stochastic game $(\mathcal{S}, \mathcal{A}^1, \ldots, \mathcal{A}^n, \mathcal{P}, \mathcal{R}^1, \ldots, \mathcal{R}^n, \gamma)$, where $\mathcal{S}$ is the state space, $\mathcal{A}^i$ is the action space of agent $i \in [1, \ldots, n]$, $\mathcal{A} = \prod_{i=1}^{n} \mathcal{A}^i$ is the joint action space of agents, $\mathcal{P} : \mathcal{S} \times \mathcal{A} \times \mathcal{S} \to [0, 1]$ is a transition function, $\mathcal{R}^i : \mathcal{S} \times \mathcal{A} \to \mathbb{R}$ is the reward function of agent $i$ , and $\gamma$ is the discount factor. The policy of agent $i$ is $\pi^i$, and the joint policy of other agents is $\pi^{-i}(a^{-i}|s) = \prod_{j \neq i} \pi^j(a^j|s)$, where $a^{-i}$ is the joint action except agent $i$. All agents interact with the environment simultaneously without communication. The historical trajectory is available, *i.e.*, for agent $i$ at timestep $t$, $\{s_0, a_0^i, a_0^{-i}, \ldots, s_{t-1}, a_{t-1}^i, a_{t-1}^{-i}\}$ is observable. The goal of the agent $i$ is to maximize its expected cumulative discount rewards

$$\mathbb{E}_{\substack{s_{t+1} \sim \mathcal{P}(\cdot|s_t, a_t^i, a_t^{-i}), \\ a^i \sim \pi^i(\cdot|s_t), a_t^{-i} \sim \pi^{-i}(\cdot|s_t)}} \left[ \sum_{t=0}^{\infty} \gamma^t \mathcal{R}^i(s_t, a_t^i, a_t^{-i}) \right]. \tag{1}$$

For convenience, the learning agent treats all other agents as a joint opponent with the joint action $a^o \sim \pi^o(\cdot|s)$ and reward $r^o$. The action and reward of the learning agent are denoted as $a \sim \pi(\cdot|s)$ and $r$, respectively.

## 4 Method

MBOM employs the environment model to predict and capture the learning of opponent policy. By simulating recursive reasoning via the environment model, MBOM imagines the learning and reasoning of the opponent and generates a set of opponent models. To obtain a stronger representation ability and accurately capture the adaptation of the opponent, MBOM mixes the imagined opponent policies according to the similarity with the real behaviors of the opponent.

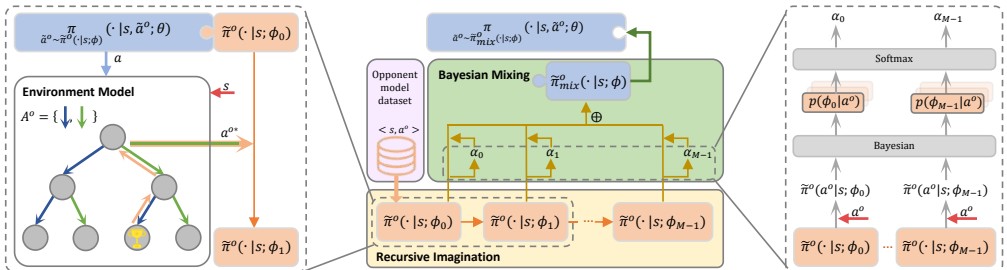

Figure 1: Architecture of MBOM

## 4.1 Recursive Imagination

If the opponent is also learning during the interaction, the opponent model fitted with historical experiences always lags behind, making the agent hard to adapt to the opponent. Moreover, if the opponent could adjust its policy according to the actions, intentions, or goals of the agent, then recursive reasoning may occur between agent and opponent. However, based on the lagged opponent model, the agent would struggle to keep up with the learning of opponent. To adapt to the learning and reasoning opponent, the agent should predict the current opponent policy and reason more deeply than the opponent.

MBOM explicitly simulates the recursive reasoning process utilizing the environment model, called *recursive imagination*, to generate a series of opponent models, called Imagined Opponent Policies (IOPs). First, we pretrain the agent's policy $\pi$ using PPO [32] while interacting with $\nu$ different opponents with diverse policies that can be learned for example by [37]. and collects a buffer $\mathcal{D}$ which contains the experience $\langle s, a, a^o, s', \boldsymbol{r}\rangle$, where $\boldsymbol{r} = \langle r, r^o\rangle$. For zero-sum game ($r + r^o = 0$) and fully cooperative game ($r = r^o$), $r^o$ can be easily obtained, while for general-sum game we make an assumption that $r^o$ can be accessed during experience collection. Then, using the experience buffer $\mathcal{D}$, we can train the environment model $\Gamma(s', \boldsymbol{r}|s, a, a^o; \zeta)$ by minimizing the mean square error

$$\mathbb{E}_{s,a,a^o,s',\boldsymbol{r}\sim\mathcal{D}} \frac{1}{2}\|s' - \hat{s}'\|^2 + \frac{1}{2}\|\boldsymbol{r} - \hat{\boldsymbol{r}}\|^2, \tag{2}$$

where $\hat{s}', \hat{\boldsymbol{r}} = \Gamma(s, a, a^o; \zeta)$, and obtain the level-0 IOP $\tilde{\pi}_0^o(\cdot|s; \phi_0)$ by maximum-likelihood estimation,

$$\mathbb{E}_{s,a^o\sim\mathcal{D}} \log \tilde{\pi}_0^o(a^o|s; \phi_0). \tag{3}$$

To imagine the learning of the opponent, as illustrated in Figure 1, we use the rollout algorithm [38] to get the best response of the opponent to the agent policy $\pi$. For each opponent action $a_t^o$ at timestep $t$, we uniformly sample the opponent action sequences in the next $k$ timesteps, simulate the trajectories using the learned environment model $\Gamma_\zeta$, and select the best response with the highest rollout value

$$a_t^{o*} = \operatorname*{argmax}_{a_t^o} \max_{a_{t+1}^o, \cdots, a_{t+k}^o \sim \text{Unif}} \sum_{j=0}^{k} \gamma^j r_{t+j}^o. \tag{4}$$

During the simulation, the agent acts according to the policy conditioned on the modeled opponent policy, $a_t \sim \pi(\cdot|s_t, \tilde{a}_t^o \sim \tilde{\pi}_0^o(\cdot|s_t; \phi_0); \theta)$, and the learned environment model provides the transition $s_{t+1}, \boldsymbol{r_t} = \Gamma(s_t, a_t, a_t^o; \zeta)$. With larger $k$, the rollout has a longer planning horizon, and thus could evaluate the action $a^{o*}$ more accurately, assuming a perfect environmental model. However, the computation cost of rollout increases exponentially with the planning horizon to get an accurate estimate of $a^{o*}$, while in practice the compounding error of the environmental model also increases with the planning horizon [15]. Therefore, the choice of $k$ is a tradeoff between accuracy and cost. Specifically, for zero-sum game and fully cooperative game, we can approximately estimate the opponent state value $V^o(s)$ as $-V(s)$ and $V(s)$, respectively, and modify the rollout value like $n$-step return [36] to obtain a longer horizon (see Appendix C for the empirical effect of $k$),

$$a_t^{o*} = \operatorname*{argmax}_{a_t^o} \max_{a_{t+1}^o, \cdots, a_{t+k}^o \sim \text{Unif}} \left[ \sum_{j=0}^{k} \gamma^j r_{t+j}^o + \gamma^{k+1} V^o(s_{t+k+1}) \right]. \tag{5}$$

---

**Algorithm 1** MBOM

1: *Pretraining:*
2: Initialize the recursive imagination layer $M$.
3: Initialize the weights $\boldsymbol{\alpha}$ with $(1/M, 1/M, \ldots, 1/M)$.
4: Interact with $\nu$ learning opponents to train the agent policy $\theta$ and collect the experience buffer $\mathcal{D}$.
5: Train the level-0 IOP $\phi_0$, the environment model $\Gamma_\zeta$ using the experience buffer $\mathcal{D}$.
6: *Interaction:*
7: **for** $t = 1, \ldots,$ max_epoch **do**
8:     Interact with the opponent to observe the real opponent actions.
    {*//Recursive Imagination*}
9:     Finetune $\phi_0$ with the real opponent actions
10:     **for** $m = 1, \ldots, M - 1$ **do**
11:         Rollout in $\Gamma_\zeta$ with $\pi(\cdot|s, \tilde{a}^o \sim \tilde{\pi}_{m-1}^o(\cdot|s; \phi_{m-1}); \theta)$ to get $a^{o*}$ by (4) or (5).
12:         Finetune $\phi_{m-1}$ with best responses to get the $\phi_m$.
13:     **end for**
    {*//Bayesian Mixing*}
14:     Update $\boldsymbol{\alpha}$ by (8).
15:     Mix the IOPs to get $\tilde{\pi}_{\text{mix}}^o$ by (6).
16: **end for**

---

By imagination, we can obtain the best response of the opponent to the agent policy $\pi$ and construct the simulated data $\{\langle s, a^{o*} \rangle\}$. Then, we use the data to finetune the level-0 IOP $\tilde{\pi}_0^o(\cdot|s; \phi_0)$ by maximum-likelihood estimation, and obtain the level-1 IOP $\tilde{\pi}_1^o(\cdot|s; \phi_1)$. The level-1 IOP can be seen as the best *response* of the opponent to the *response* of the agent conditioned on level-0 IOP. In the imagination, it is the nested form as "the opponent believes [that the agent believes (that the opponent believes ...)]." The existing imagined opponent policy is the innermost "(that the opponent believes)," while the outermost "the opponent believes" is $a^{o*}$ which is obtained by the rollout process. Recursively repeating the rollout and finetuning, where the agent policy is conditioned on the IOP $\tilde{\pi}_{m-1}^o(\cdot|s; \phi_{m-1})$, we could derive the level-$m$ IOP $\tilde{\pi}_m^o(\cdot|s; \phi_m)$.

## 4.2 Bayesian Mixing

By recursive imagination, we get $M$ IOPs with different reasoning levels. However, since the learning and reasoning of the opponent are unknown, a single IOP might erroneously estimate the opponent. To obtain stronger representation capability and accurately capture the learning of the opponent, we linearly combine the IOPs to get a mixed IOP,

$$\tilde{\pi}_{\text{mix}}^o(\cdot|s) = \sum_{m=0}^{M-1} \alpha_m \tilde{\pi}_m^o(\cdot|s; \phi_m), \tag{6}$$

where $\alpha_m$ is the weight of level-$m$ IOP, which is taken as $p(\tilde{\pi}_m^o|a^o)$, also denoted as $p(m|a^o)$, the probability that the opponent action $a^o$ is generated from the level-$m$ IOP. Thus, $p(m|a^o)$ indicates the similarity between the level-$m$ IOP $\tilde{\pi}_m^o$ and the opponent policy $\pi^o$ in the most recent stage. By Bayesian rule, we have

$$p(m|a^o) = \frac{p(a^o|m)p(m)}{\sum_{i=0}^{M-1} p(a^o|i)p(i)} = \frac{\tilde{\pi}_m^o(a^o|s; \phi_m)p(m)}{\sum_{i=0}^{M-1} [\tilde{\pi}_i^o(a^o|s; \phi_i)p(i)]}. \tag{7}$$

Updating $p(m|a^o)$ during interaction can obtain a more accurate estimate of the improving opponent policy.

In practice, we use the moving average of $p(m|a^o)$ as the prior $p(m)$, take the decayed moving average of $p(m|a^o)$ as $\Psi_m$, and obtain $\boldsymbol{\alpha}$ by the IOPs mixer,

$$\boldsymbol{\alpha} = (\alpha_0, \ldots, \alpha_{M-1}) = \text{softer-softmax}(\Psi_0, \ldots, \Psi_{M-1}). \tag{8}$$

Softer-softmax [12] is a variant of softmax, which uses higher temperature to control a softy of the probability distribution. The IOPs mixer is non-parametric, which could be updated quickly and efficiently without parameter training and too many interactions (see Appendix B for empirical analysis on $\boldsymbol{\alpha}$). Therefore, the IOPs mixer could adapt to the fast-improving opponent.

For completeness, the full procedure of MBOM in given in Algorithm 1.

### 4.3 Theoretical Analysis

MBOM consists of several components, including a conditional policy, an environment model, and $M$ IOPs. They all interact with each other through model rollouts, which however makes it extremely hard to give an error analysis based on each component of MBOM. Therefore, we instead focus mainly on recursive imagination and Bayesian mixing, and give a profound understanding of MBOM. Proofs are available in Appendix A.

Without loss of generality, we define $\delta_m$ as the discrepancy between level-$m$ and $m$-1 IOPs in terms of estimated value function given the IOP, and $\varepsilon_m$ as the error of level-$m$ IOP compared to the true value function given the opponent policy, *i.e.*, $\delta_m \doteq |\widehat{V}_m - \widehat{V}_{m-1}|$, $\varepsilon_m \doteq |\widehat{V}_m - V|$, and $\delta_0 \doteq \varepsilon_0$. Then, we have the following two lemmas.

**Lemma 1.** *For the mixed imagined opponent policy (IOP) $\tilde{\pi}^o_{\mathrm{mix}}(\cdot|s) = \sum_{m=0}^{M-1} \alpha_m \tilde{\pi}^o_m(\cdot|s; \phi_m)$, the total error $\varepsilon_{total} = \sum_{m=0}^{M-1} \alpha_m \varepsilon_m$ satisfies the following inequality:*

$$\varepsilon_{total} \leq \sum_{m=0}^{M-1} \delta_m \sum_{i=m}^{M-1} \alpha_i.$$

**Lemma 2.** *Suppose the value function is Lipschitz continuous on the state space $\mathcal{S}$, $K$ is the Lipschitz constant, $\widehat{\mathcal{M}}_i(s,a) \doteq \Gamma(s,a,a^o;\zeta) \cdot \tilde{\pi}^o_i(a^o|s;\phi_i)$ is the transition distribution given $s$ and $a$, then*

$$\left|\widehat{V}_i - \widehat{V}_j\right| \leq \frac{\gamma K}{1-\gamma} \cdot \mathop{\mathbb{E}}_{s,a \sim \pi, \widehat{\mathcal{M}}_j} \left\|\widehat{\mathcal{M}}_i(s,a) - \widehat{\mathcal{M}}_j(s,a)\right\|.$$

According to (6) and (7), we can express $\alpha_m$ as the following

$$\alpha_m \doteq \mathbb{E}\left[p(m|a^o)\right] = \sum_{s,a^o} d(s,a^o) \frac{\tilde{\pi}^o_m(a^o|s;\phi_m)\, p(m)}{\sum_{i=0}^{M-1} \tilde{\pi}^o_i(a^o|s;\phi_i)\, p(i)},$$

where $d(s,a^o)$ is the stationary distribution of pairs of state and action of opponent. Then, based on Lemma 1 and 2, we can obtain the following theorem, where $\widehat{\mathcal{M}}_{-1}$ denotes $\mathcal{P}(\cdot|s,a,a^o) \cdot \pi^o(a^o|s)$, the true transition dynamics of agent.

**Theorem 1.** *Define the error between the approximated value function of the Bayesian mixing IOP and the true value function as $\varepsilon_{true} \doteq \left|\widehat{V} - V\right|$. For the Bayesian mixing IOP, the true error satisfies the following inequality:*

$$\varepsilon_{true} \leq \varepsilon_{total} \leq \sum_{s,a^o} \frac{\gamma K d(s,a^o)}{1-\gamma} \cdot \frac{\sum_{m=0}^{M-1} \mathop{\mathbb{E}}_{s,a \sim \pi, \widehat{\mathcal{M}}_{m-1}} \left\|\widehat{\mathcal{M}}_m - \widehat{\mathcal{M}}_{m-1}\right\| \sum_{i=m}^{M-1} \tilde{\pi}^o_i(a^o|s;\phi_i)\, p(i)}{\sum_{j=0}^{M-1} \tilde{\pi}^o_j(a^o|s;\phi_j)\, p(j)}.$$

Theorem 1 tells us the error accumulates as level $M$ increases. However, larger $M$ also has advantages. To analyze, we first define the benefit using the mixed IOP as In the error bound of Theorem 1, it is easy to see that the error accumulates as level $M$ increases, but at the same time, higher level also has advantages. To analyze, we need to define the benefit of using the mixed IOP as the optimization. We use the negative distribution distance $\eta \doteq -\|\hat{\pi} - \pi\|$ to denote the benefit of using policy in different levels, where $\hat{\pi}$ is the estimated opponent policy, $\pi$ is the true opponent policy and $\|\cdot\|$ is a general form of distribution distance function. For a mixing policy, we can define the benefit as:

$$\eta_M = -\|(\alpha_0 \tilde{\pi}^o_0 + \cdots + \alpha_{M-1} \tilde{\pi}^o_{M-1}) - \pi^o\|.$$

Then, we have the following lemma and theorem.

**Lemma 3.** *Assume that the true opponent policy $\pi^o$ has a probability distribution over the given set of IOPs $\{\tilde{\pi}^o_0, \ldots, \tilde{\pi}^o_{M-1}\}$, and a probability $\Pr\{\pi^o = \tilde{\pi}^o_m\} = p_m$ for each $\tilde{\pi}^o_m$, then the maximum expectation of $\eta_M$ can be achieved if and only if $\alpha_m = p_m, \forall m \in [0, M-1]$.*

**Theorem 2.** *Given the action trajectory of opponent $\{a_0^o, a_1^o, \ldots, a_t^o\}$, the posterior probability updated by Bayesian mixing approximates the true probability of opponent as the length of the trajectory grows, i.e.,*

$$P(m|a_t^o, \cdots, a_1^o, a_0^o) \to P_{\text{true}}(m) \doteq p_m, \quad t \to \infty.$$

*Then the maximum expectation of $\eta_M$ can be achieved with $\alpha_m = \mathbb{E}[P(m|a_t^o, \cdots, a_1^o, a_0^o)]$.*

According to Theorem 2, we know that the estimated probability distribution of the opponent converges to the true distribution of the opponent. It is obvious that larger $M$ improves the representation capability of IOPs and thus better satisfies the assumption in Lemma 3, but also increases the error bound in Theorem 1. Therefore, the selection of $M$ is a tradeoff between them (see Appendix C for empirically study on $M$).

## 5 Experiments

We first evaluate MBOM thoroughly in two-player zero-sum tasks. Then, we investigate MBOM when against multiple opponents and in a cooperative task. In all the experiments, the baselines have the same neural network architectures as MBOM. All the methods are trained for five runs with different random seeds, and results are presented using mean and 95% confidence intervals. More details about experimental settings and hyperparameters are available in Appendix E.

### 5.1 Two-Player Zero-Sum Tasks

We evaluate MBOM in two competitive tasks: (1) **Triangle Game** is an asymmetric zero-sum game implemented on Multi-Agent Particle Environments (MPE) [21]. When facing different policies of the opponent, the agent has to adjust its policy to adapt to the opponent for higher reward. (2) **One-on-One** is a two-player competitive game implemented on Google Research Football [18]. The goalkeeper controlled by the agent could only passively react to the strategies of the shooter controlled by the opponent and makes policy adaptation when the shooter strategy changes.

**Baselines.** In the experiments, we compare MBOM with the following methods:

- LOLA-DiCE [9] is an expansion of the LOLA, which uses Differentiable Monte-Carlo Estimator (DiCE) operation to consider how to shape the learning dynamics of other agents.
- Meta-PG [1] uses trajectories from the current opponents to do multiple meta-gradient steps and construct a policy that is good for the updated opponents.
- Meta-MAPG [16] includes an additional term that accounts for the impact of the agent's current policy on the future policies of opponents, compared with Meta-PG.
- PPO [32] is a classical single-agent RL algorithm, without any other modules.

**Opponents.** We construct three types of opponents:

- Fixed policy. The opponents are pre-trained but not updated during interaction.
- Naïve learner. The opponents are pre-trained and updated using PPO during interaction.
- Reasoning learner. The opponents are pre-trained and can model the behavior of the agent. The model is finetuned during interaction and their policy is conditioned on the predicted action of the model.

**Performance.** The experimental results against test opponents in Triangle Game and One-on-One are shown in Figure 2, and the mean performance with standard deviation over all test opponents is summarized in Table 1. Without explicitly considering opponent policy, PPO achieves poor performance. The learning of LOLA-DiCE depends on the gradient of the opponent model. However, the gradient information cannot clearly reflect the distinctions between diverse opponents, which leads to that LOLA-DiCE cannot adapt to the unseen opponent quickly and effectively. Meta-PG and Meta-MAPG show mediocre performance in Triangle Game. In One-on-One, since Meta-PG and Meta-MAPG heavily rely on the reward signal, adaptation is difficult for the two methods in this sparse reward task. MBOM outperforms Meta-MAPG and Meta-PG, because the opponent model of MBOM improves the ability to adapt to different opponents. And the performance gain becomes more significant when the opponent is naïve learner or reasoning learner, which is attributed

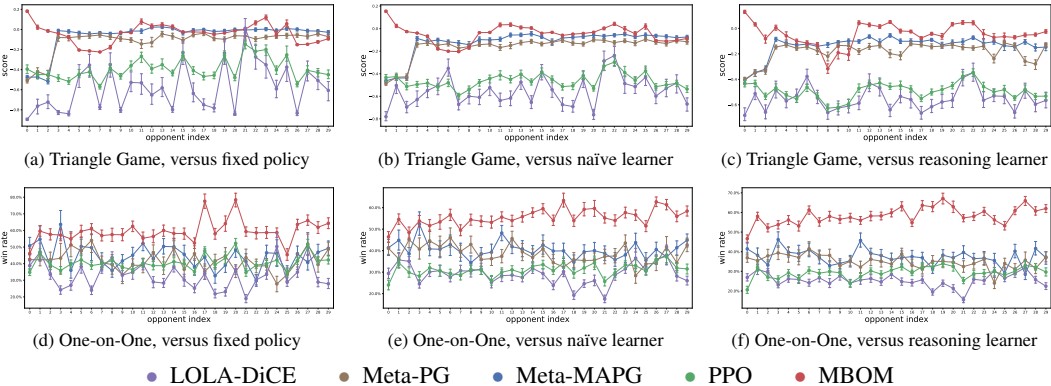

(a) Triangle Game, versus fixed policy     (b) Triangle Game, versus naïve learner     (c) Triangle Game, versus reasoning learner

(d) One-on-One, versus fixed policy     (e) One-on-One, versus naïve learner     (f) One-on-One, versus reasoning learner

● LOLA-DiCE    ● Meta-PG    ● Meta-MAPG    ● PPO    ● MBOM

Figure 2: Performance against different types of opponents, *i.e.*, fixed policy, naïve learner, and reasoning learner, where *x*-axis is opponent index. The results show that MBOM outperforms other baselines, especially against naïve learner and reasoning learner.

to the recursive reasoning capability brought by recursive imagination in the environment model and Bayesian mixing that quickly captures the learning of the opponent.

Table 1: Performance on Triangle Game and One-on-One

| Methods | Triangle Game (score ↑) | | | One-on-One (win rate %) | | |
|---|---|---|---|---|---|---|
| | Fixed Policy | Naïve Learner | Reasoning Learner | Fixed Policy | Naïve Learner | Reasoning Learner |
| LOLA-DiCE | $-22.51\,(5.22)$ | $-20.48\,(4.02)$ | $-21.55\,(4.43)$ | $33.0\,(0.5)$ | $25.2\,(1.2)$ | $18.4\,(0.9)$ |
| Meta-PG | $-3.78\,(0.13)$ | $-6.72\,(0.18)$ | $-8.35\,(1.87)$ | $41.7\,(0.7)$ | $35.4\,(2.5)$ | $28.0\,(0.6)$ |
| Meta-MAPG | $-2.01\,(0.06)$ | $-5.76\,(0.29)$ | $-6.14\,(0.84)$ | $44.4\,(1.5)$ | $36.6\,(1.6)$ | $31.1\,(1.9)$ |
| PPO | $-13.29\,(1.80)$ | $-18.42\,(1.28)$ | $-20.51\,(1.80)$ | $40.6\,(0.7)$ | $21.5\,(0.7)$ | $25.8\,(1.0)$ |
| MBOM | $\mathbf{-1.19\,(0.03)}$ | $\mathbf{-1.66\,(0.29)}$ | $\mathbf{-2.75\,(0.89)}$ | $\mathbf{59.5\,(0.9)}$ | $\mathbf{51.3\,(1.0)}$ | $\mathbf{64.2\,(1.8)}$ |

**Ablation Studies.** The opponent modeling module of MBOM is composed of both recursive imagination and Bayesian mixing. In the following, we respectively test the functionality of these two components.

In *recursive imagination*, MBOM generates a sequence of different levels of IOPs finetuned by the imagined best response of the opponent in the environment model. For comparison, we use only $\phi_0$ as the opponent model (*i.e.*, without recursive imagination), denoted as MBOM w/o IOPs, and use random actions to finetune the IOPs rather than using the best response, denoted as MBOM-BM. Note that MBOM w/o IOPs is essentially the same as the simple opponent modeling methods that predict opponents' actions, like [11, 13, 10]. The experimental results of MBOM, MBOM w/o IOPs, and MBOM-BM are shown in Figure 3(a) and 3(b). MBOM w/o IOPs obtains similar results to MBOM when facing fixed policy opponents because $\phi_0$ could accurately predict the opponent behaviors if the opponent is fixed, which corroborates the empirical results in [11, 13, 10]. However, if the opponent is learning or reasoning, $\phi_0$ cannot accurately represent the opponent, thus the performance of MBOM w/o IOPs drops. The methods with IOPs perform better than MBOM w/o IOPs, which means that a variety of IOPs can capture the policy changes of learning opponents. When facing the reasoning learner, MBOM outperforms MBOM-BM, indicating that the IOPs finetuned by the imagined best response have a stronger ability to represent the reasoning learner.

In *Bayesian mixing*, MBOM mixes the generated IOPs to obtain a policy that is close to the real opponent. As a comparison, we directly use the individual level-$m$ IOP generated by recursive imagination without mixing, denoted as MBOM-$\phi_m$, and use uniformly mixing instead of Bayesian mixing, denoted as MBOM-unif. The experimental results are illustrated in Figure 3(c) and 3(d). Benefited from recursive imagination, MBOM-$\phi_1$ and MBOM-$\phi_2$ show stronger performance than MBOM-$\phi_0$. MBOM consistently outperforms the ablation baselines without mixing when fighting against reasoning learners. The middling performance of MBOM-unif is due to not exploiting the more representative IOPs, indicating that Bayesian mixing could obtain a more accurate estimate of reasoning opponents.

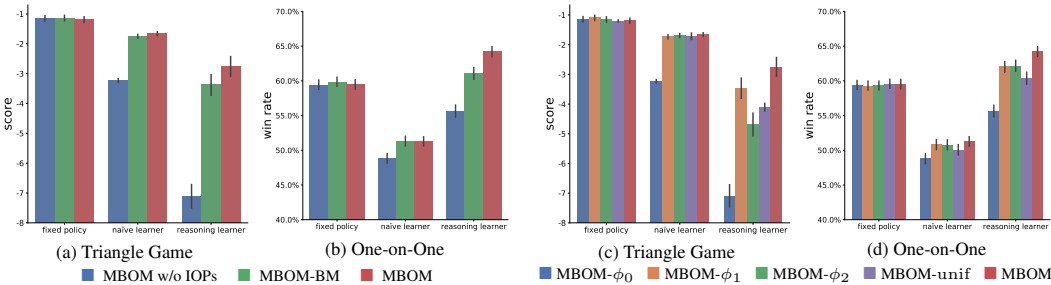

Figure 3: Ablation study of MBOM. MBOM is compared with MBOM-BM and MBOM w/o IOPs on *recursive imagination* in (a) Triangle Game and (b) One-on-One. MBOM is compared with MBOM-$\phi_0$, MBOM-$\phi_1$, MBOM-$\phi_2$, and MBOM-unif on *Bayesian mixing* in (c) Triangle Game and (d) One-on-One.

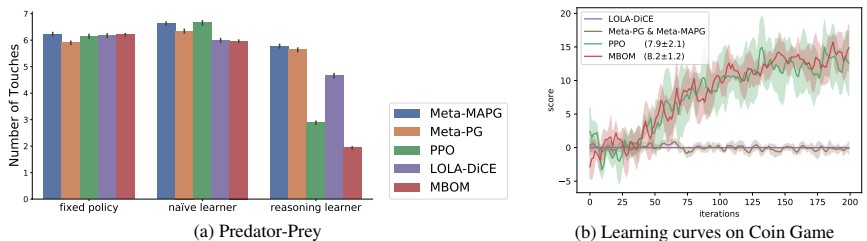

Figure 4: (a) Performance against different types of opponents in Predator-Prey, where smaller touch number means better performance. (b) Learning curves in Coin Game, where the numbers in the legend are the mean score with standard deviation over 200 iterations.

We additionally provide analysis on the weight $\alpha$ and ablation studies on hyperparameters including the rollout length $k$ and the level of recursive imagination $M$, which are available in Appendix B and C, respectively.

## 5.2  Multiple Opponents

When facing multiple opponents, MBOM takes them as a joint opponent, which is consistent with the single-opponent case. We perform experiments in Predator-Prey [21]. We control the prey as the agent and take the three predators as opponents. In this task, the agent tries not to be touched by the three opponents. When the policies of opponents are fixed (*e.g.*, enveloping the agent), it is difficult for the agent to get high reward. However, if the policies of opponents change during interaction, there may be changes for the agent to escape. The results are shown in Figure 4(a). When facing learning opponents, MBOM achieves performance gain, especially competing with the reasoning learner, which indicates that MBOM adapts to the learning and reasoning of the opponents and responds effectively. Meta-MAPG and Meta-PG do not effectively adapt to the changes of opponents' policies and are prone to be touched by the opponents many times in a small area, resulting in poor performance. LOLA-DiCE, Meta-PG and Meta-MAPG underperforms PPO when against multiple reasoning learners, indicating their adaptation may induce a negative affect on the agent performance. Detailed experimental results with different opponents are available in Appendix D.

## 5.3  Cooperative Task

MBOM could also be applied to cooperative tasks. We test MBOM on a cooperative scenario, Coin Game, which is a high-dimension expansion of the iterated prisoner dilemma with multi-step actions [20, 8]. Both agents simultaneously update their policies using MBOM or the baseline methods to maximize the sum of rewards. The experiment results are shown in Figure 4(b). Meta-PG and Meta-MAPG degenerate to Policy Gradients for this task as there is no training set. Both learn a greedy strategy that collects any color coin, which leads to a zero total score of two players. LOLA-DiCE learns too slow and does not learn to cooperate within 200 iterations, indicating the inefficiency of estimating the opponent gradients. PPO learns to cooperate quickly and successfully,

which corroborates the good performance of independent PPO in cooperative multi-agent tasks as pointed out in [6, 44]. MBOM slightly outperforms PPO, which indicates that MBOM can also be applied to cooperative tasks without negative effects. Note that we do not compare other cooperative MARL methods like QMIX [31], as they require centralized training.

### 5.4 Opponent Model Accuracy Analysis

MBOM intends to achieve a higher return by improving model accuracy via adapting to the changing opponent. In previous sections, we have shown that MBOM achieves superior performance. In this section, we investigate the correlation between the performance and the prediction accuracy of the opponent model.

First, we tested the prediction error of the opponent model, using KL divergence to measure the distance between the predicted action distribution and the real action distribution of the opponent policy. The results compared with MBOM w/o IOPs (without recursive imagination) are shown in Table 2 (the column *Error*). Overall, MBOM's opponent model achieves a more accurate prediction than MBOM w/o IOPs.

Then, we tested the performance of the same policy $\pi(\cdot|s, \tilde{a}^o; \theta)$ with different opponent models. The opponent model of MBOM w/o IOPs is continuously updated by supervised learning using the observed opponent's action, which has lower prediction accuracy than MBOM's opponent model. MBOM-pro, which uses the true actions of the opponent as the input of the policy during the interaction phase instead of the actions predicted by IOPs. That is, it can be considered that the prediction accuracy of MBOM-pro's opponent model is 100%. With the increasing prediction accuracy of the opponent model, the performance also increases correspondingly, from MBOM w/o IOPs → MBOM → MBOM-pro, as shown in Table 2 (the column *Performance*). The results show that a more accurate opponent model could improve performance, and that MBOM indeed obtains a more accurate opponent model when the opponent is learning.

Table 2: Prediction accuracy of opponent model and performance in Triangle Game and One-on-One.

| | | Fixed policy | | Naïve learner | | Reasoning learner | |
|---|---|---|---|---|---|---|---|
| | | Error ↓ | Performance | Error ↓ | Performance | Error ↓ | Performance |
| Triangle Game | MBOM w/o IOPs | 15.72 (0.18) | −1.14 (0.03) | 15.17 (0.17) | −3.23 (0.07) | 8.19 (0.09) | −7.10 (2.00) |
| | MBOM | 15.47 (0.04) | −1.19 (0.03) | 13.47 (0.43) | −1.66 (0.29) | 7.81 (0.38) | −2.75 (0.89) |
| | MBOM-pro | - | 0.96 (0.03) | - | −0.11 (0.18) | - | 0.82 (0.35) |
| One-on-One | MBOM w/o IOPs | 5.62 (0.06) | 59.4 (0.7) | 6.04 (0.14) | 48.8 (1.7) | 7.36 (0.34) | 55.7 (1.9) |
| | MBOM | 5.78 (0.05) | 59.5 (0.9) | 5.78 (0.06) | 51.3 (1.0) | 7.04 (0.12) | 64.2 (1.8) |
| | MBOM-pro | - | 59.8 (0.6) | - | 52.0 (1.7) | - | 67.5 (0.4) |

## 6 Conclusion and Future Work

We have proposed MBOM, which employs recursive imagination and Bayesian mixing to predict and capture the learning and improvement of opponents. Empirically, we evaluated MBOM in a variety of competitive tasks and demonstrated MBOM adapts to learning and reasoning opponents much better than the baselines. These make MBOM a simple and effective RL method whether opponents be fixed, continuously learning, or reasoning in competitive environments. Moreover, MBOM can also be applied in cooperative tasks.

In MBOM, the learning agent treats all opponents as a joint opponent. If the size of the joint opponent is large, the agent will need a lot of rollouts to get an accurate best response. The cost increases dramatically with the size of the joint opponent. How to reduce the computation overhead in such scenarios will be considered in future work. Moreover, MBOM implicitly assumes that the relationship between opponents is fully cooperative. How to deal with the case where their relationship is non-cooperative is also left as future work.

### Acknowledgments and Disclosure of Funding

This work was supported in part by NSF China under grant 61872009. The authors would like to thank the anonymous reviewers for their valuable comments.

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
