# A Proofs

**Lemma 1.** *For the mixed imaged opponent policy (IOP) $\tilde{\pi}_{\text{mix}}^o(\cdot|s) = \sum\limits_{m=0}^{M-1} \alpha_m \tilde{\pi}_m^o(\cdot|s; \phi_m)$, the total error satisfies the following inequality:*

$$\varepsilon_{total} \leq \sum_{m=0}^{M-1} \delta_m \sum_{i=m}^{M-1} \alpha_i.$$

*Proof.* By the definitions of $\delta_m$ and $\varepsilon_m$, it is clear that

$$\varepsilon_m = |\widehat{V}_m - V| \leq |\widehat{V}_m - \widehat{V}_{m-1}| + \cdots + |\widehat{V}_1 - \widehat{V}_0| + |\widehat{V}_0 - V|$$
$$= \delta_m + \cdots + \delta_1 + \delta_0$$

Then, we can derive the following inequality,

$$\varepsilon_{total} = \sum_{m=0}^{M-1} \alpha_m \varepsilon_m \leq \sum_{m=0}^{M-1} \alpha_m \sum_{k=0}^{m} \delta_k$$
$$= \sum_{m=0}^{M-1} \delta_m \sum_{i=m}^{M-1} \alpha_i$$

$\square$

**Lemma 2.** *Suppose the value function is Lipschitz continuous on the state space $\mathcal{S}$, $K$ is the Lipschitz constant, $\widehat{\mathcal{M}}_i(s,a) \doteq \Gamma(s, a, a^o; \zeta) \cdot \tilde{\pi}_i^o(a^o|s; \phi_i)$ is the transition distribution given $s$ and $a$, then*

$$\left|\widehat{V}_i - \widehat{V}_j\right| \leq \frac{\gamma K}{1 - \gamma} \cdot \mathop{\mathbb{E}}_{s,a\sim\pi,\widehat{\mathcal{M}}_j} \left\|\widehat{\mathcal{M}}_i(s,a) - \widehat{\mathcal{M}}_j(s,a)\right\|.$$

*Proof.* Lemma 2 is a directly cited theorem in [22], and we make some modifications to fit our context. $\square$

**Theorem 1.** *Define the error between the approximated value function of the Bayesian mixing IOP and the true value function as $\varepsilon_{true} \doteq \left|\widehat{V} - V\right|$. For the Bayesian mixing IOP, the true error satisfies the following inequality:*

$$\varepsilon_{true} \leq \varepsilon_{total} \leq \sum_{s,a^o} \frac{\gamma K d(s, a^o)}{1 - \gamma} \cdot \frac{\sum\limits_{m=0}^{M-1} \mathop{\mathbb{E}}_{s,a\sim\pi,\widehat{\mathcal{M}}_{m-1}} \left\|\widehat{\mathcal{M}}_m - \widehat{\mathcal{M}}_{m-1}\right\| \sum\limits_{i=m}^{M-1} \tilde{\pi}_i^o\left(a^o|s; \phi_i\right) p(i)}{\sum\limits_{j=0}^{M-1} \tilde{\pi}_j^o\left(a^o|s; \phi_j\right) p(j)}.$$

*Proof.* It is clear that

$$\varepsilon_{true} \doteq \left|\widehat{V} - V\right| = \left|\sum_{m=0}^{M-1} \alpha_m \widehat{V}_m - V\right|$$
$$\leq \sum_{m=0}^{M-1} \alpha_m \left|\widehat{V}_m - V\right|$$
$$= \sum_{m=0}^{M-1} \alpha_m \varepsilon_m = \varepsilon_{total}$$

Then, we can derive the following inequality,

$$\varepsilon_{true} \leq \varepsilon_{total} \leq \sum_{m=0}^{M-1} \delta_m \sum_{i=m}^{M-1} \alpha_i = \sum_{m=0}^{M-1} \delta_m \sum_{i=m}^{M-1} \sum_{s,a^o} d(s,a^o) \frac{\tilde{\pi}_i^o\left(a^o|s;\phi_i\right) p(i)}{\sum\limits_{j=0}^{M-1} \tilde{\pi}_j^o\left(a^o|s;\phi_j\right) p(j)}$$

$$= \sum_{s,a^o} d(s,a^o) \frac{\sum\limits_{m=0}^{M-1} \delta_m \sum\limits_{i=m}^{M-1} \tilde{\pi}_i^o\left(a^o|s;\phi_i\right) p(i)}{\sum\limits_{j=0}^{M-1} \tilde{\pi}_j^o\left(a^o|s;\phi_j\right) p(j)}$$

$$= \sum_{s,a^o} \frac{\gamma K d(s,a^o)}{1-\gamma} \cdot \frac{\sum\limits_{m=0}^{M-1} \mathop{\mathbb{E}}\limits_{s,a\sim\pi,\widehat{\mathcal{M}}_{m-1}} \left\|\widehat{\mathcal{M}}_m - \widehat{\mathcal{M}}_{m-1}\right\| \sum\limits_{i=m}^{M-1} \tilde{\pi}_i^o\left(a^o|s;\phi_i\right) p(i)}{\sum\limits_{j=0}^{M-1} \tilde{\pi}_j^o\left(a^o|s;\phi_j\right) p(j)},$$

where $d(s,a_o)$ is the stationary distribution of pairs of state and action of opponent and $\widehat{\mathcal{M}}_{-1}$ denotes $\mathcal{P}(\cdot|s,a,a^o) \cdot \pi^o(a^o|s)$. $\qquad\square$

**Lemma 3.** *Assume that the true opponent policy $\pi^o$ has a probability distribution over the given set of IOPs $\{\tilde{\pi}_0^o, \ldots, \tilde{\pi}_{M-1}^o\}$, and a probability $\Pr\{\pi^o = \tilde{\pi}_m^o\} = p_m$ for each $\tilde{\pi}_m^o$, then the maximum expectation of $\eta_M$ can be achieved if and only if $\alpha_m = p_m, \forall m \in [0, M-1]$.*

*Proof.*

$$\mathbb{E}[\eta_M] = -\mathbb{E}\|(\alpha_0\tilde{\pi}_0^o + \cdots + \alpha_{M-1}\tilde{\pi}_{M-1}^o) - \pi\|$$

$$= -\sum_{m=0}^{M-1} p_m\|(\alpha_0\tilde{\pi}_0^o + \cdots + \alpha_{M-1}\tilde{\pi}_{M-1}^o) - \tilde{\pi}_m^o\|$$

$$\geq -\sum_{m=0}^{M-1} \|\alpha_j\tilde{\pi}_m^o - p_m\tilde{\pi}_m^o\|$$

Since $\mathbb{E}[\eta_M]$ is a negative distance function, it is clear that

$$\mathbb{E}[\eta_M] \leq 0,$$

and when $\alpha_m = p_m, \forall m \in [0, M-1]$,

$$\mathbb{E}[\eta_M] \geq -\sum_{m=0}^{M-1} \|p_m\tilde{\pi}_m^o - p_m\tilde{\pi}_m^o\| = 0$$

Therefore, the maximum expectation of $\eta_M$ can be achieved if and only if $\alpha_m = p_m, \forall m \in [0, M-1]$. $\qquad\square$

**Theorem 2.** *Given the action trajectory of opponent $\{a_0^o, a_1^o, \ldots, a_t^o\}$, the posterior probability updated by Bayesian mixing approximates the true probability of opponent as the length of the trajectory grows, i.e.,*

$$P(m|a_t^o, \cdots, a_1^o, a_0^o) \to P_{\text{true}}(m), \quad t \to \infty.$$

*Then the maximum expectation of $\eta_M$ can be achieved with $\alpha_m = \mathbb{E}[P(m|a_t^o, \cdots, a_1^o, a_0^o)]$.*

*Proof.* According to Bayes' theorem, as we update the posterior probability as (7), we have the following posterior probabilities

$$p(m|a_0^o) = \frac{p(a_0^o|m)p(m)}{\sum_{i=0}^{M-1} p(a_0^o|i)p(i)}$$

$$p(m|a_1^o, a_0^o) = \frac{p(a_1^o|a_0^o, m)p(a_0^o|m)p(m)}{\left[\sum_{i=0}^{M-1} p(a_1^o|, a_0^o, i)p(i)\right] \cdot \left[\sum_{i=0}^{M-1} p(a_0^o|i)p(i)\right]}$$

$$\vdots$$

$$p(m|a_t^o, \cdots, a_0^o) = \frac{p(a_t^o|a_{t-1}^o, \cdots, a_0^o, m) \cdots p(a_0^o|m)p(m)}{\left[\sum_{i=0}^{M-1} p(a_t^o|a_{t-1}^o, \cdots, a_0^o, i)p(i)\right] \cdots \left[\sum_{i=0}^{M-1} p(a_0^o|i)p(i)\right]}$$

Obviously, with sufficient samples,

$$P(m|a_t^o, \cdots, a_0^o) \to P_{\text{true}}(m) \doteq p_m, \quad t \to \infty.$$

This can also be expressed as

$$\mathbb{E}[P(m|a_t^o, \cdots, a_1^o, a_0^o)] = p_m$$

When $\alpha_m = \mathbb{E}[P(m|a_t^o, \cdots, a_1^o, a_0^o)]$, we have $\alpha_m = p_m$. Considering Lemma 3, the maximum expectation of $\eta_M$ is achieved. $\qquad\square$

# B  Weights $\alpha$

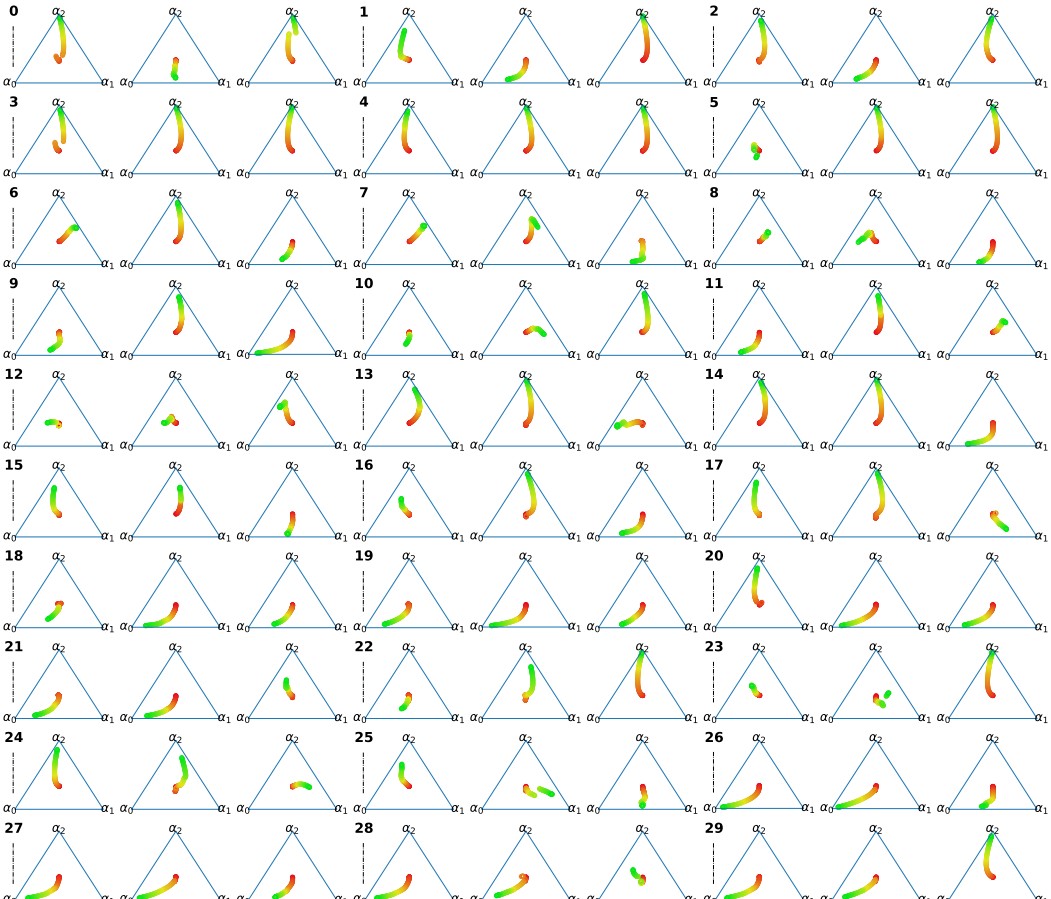

Figure 5: The change of $\boldsymbol{\alpha}$ during the adaptation, colored from red (initial $\alpha$) to green (end), in Triangle Game. In each subplot, the top left number is the opponent index, and from left to right are shown the agent plays against fixed policy, naïve learner, reasoning learner.

In Bayesian mixing, $\boldsymbol{\alpha}$ is the weights to mix IOPs,

$$\boldsymbol{\alpha} = (\alpha_0, \ldots, \alpha_{M-1}) = \text{softer-softmax}(\Psi_0, \ldots, \Psi_{M-1}).$$

$\Psi_m^t$ of level-$m$ IOP at timestep $t$ is

$$\Psi_m^t = \sum_{l=t-H}^{t-1} \lambda^{t-l} p(m|a_l^o),$$

where $\lambda$ is the decay factor, $H$ is the horizon, $p(m|a_t^o)$ can be calculated using (7). Moreover, $p(m)$ at timestep $t$ is the moving average of $p(m|a)$ over the horizon $H$ as

$$p(m) = \sum_{l=t-H}^{t-1} p(m|a_l^o)/H.$$

Figure 5 and 6 depict the change of $\boldsymbol{\alpha}$ ($M = 3$) during the adaptation when the agent plays against different test opponents in Triangle Game and One-one-One, respectively. In each subplot, the top left number is the opponent index, and from left to right are fixed policy, naïve learner, reasoning learner. The changing trends of $\boldsymbol{\alpha}$ are diverse when against different opponents.

As the level-0 IOP is finetuned during interaction, when playing against fixed policy, $\alpha_0$ should be large. When playing against reasoning learner, intuitively $\alpha_1$ should be large. However, the

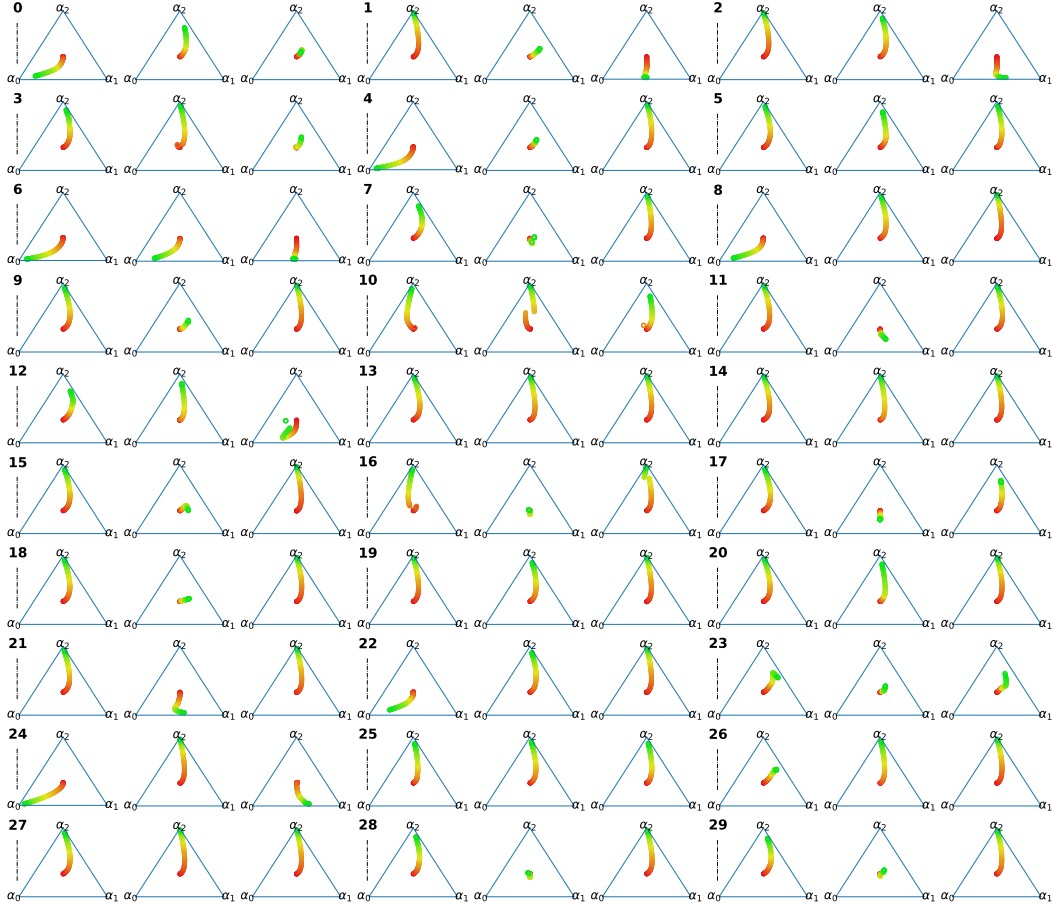

Figure 6: The change of $\alpha$ during the adaptation, colored from red (initial $\alpha$) to green (end), in One-on-One. In each subplot, the top left number is the opponent index, and from left to right are shown the agent plays against fixed policy, naïve learner, reasoning learner.

naïve learner's policy is updated only with reward, thus it does not have a counterpart in the IOPs. As illustrated in Figure 5 and 6, $\alpha$ does not always converge to the corresponding level of IOP when against fixed policy and reasoning learner. The reason is two-fold. First, as the level-0 IOP is pre-trained against training opponents that are different from test opponents (will be discussed in Appendix E), the level-0 IOP can be largely different from the opponent policy. Thus, the small number of samples obtained during online interaction may not be enough to finetune the level-0 IOP to accurately model the opponent policy. This is referred to as the error of opponent modeling, Thus, $\alpha$ does not always converge to $\alpha_0$ when testing against fixed policy. Second, due to such an inaccurate level-0 IOP and the error of the environment model, higher-level IOPs may also be inaccurate, thus $\alpha$ does not always converge to $\alpha_1$ when testing against reasoning learner. Essentially, $\alpha$ is a mapping from IOPs reasoned by the agent to the true policy generated by the opponent's learning method. When testing against different types of opponents, the mixed IOP according to $\alpha$ may already be capable to well represent the true opponent policy and capture its update, which offsets the errors of opponent modeling and environment model and makes MBOM almost intact and outperform the baselines.

# C Ablation on Hyperparameters

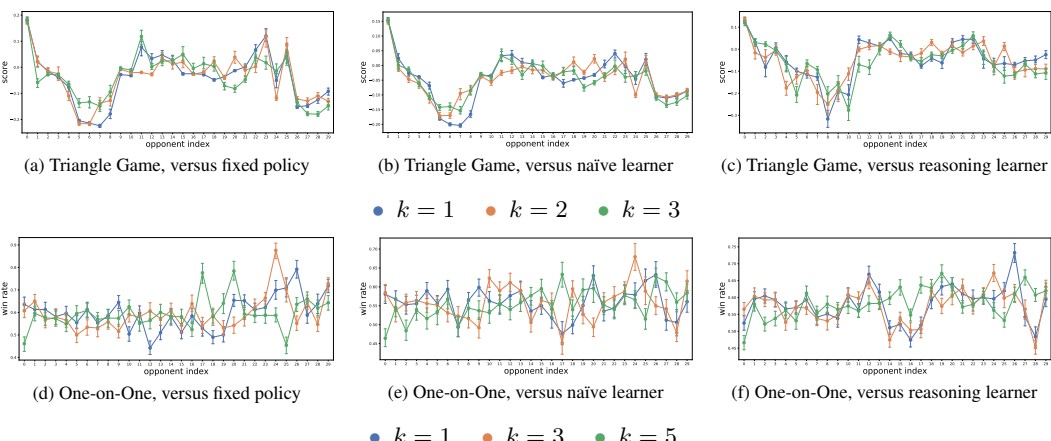

(a) Triangle Game, versus fixed policy  (b) Triangle Game, versus naïve learner  (c) Triangle Game, versus reasoning learner

● $k = 1$   ● $k = 2$   ● $k = 3$

(d) One-on-One, versus fixed policy  (e) One-on-One, versus naïve learner  (f) One-on-One, versus reasoning learner

● $k = 1$   ● $k = 3$   ● $k = 5$

Figure 7: Performance against different types of opponents, *i.e.*, fixed policy, naïve learner, and reasoning learner with different rollout planning horizon $k$ as in (4). The selection of $k$ is a tradeoff between the error of the environment model error and the estimate accuracy of rollout. The results are plotted using mean and 95% confidence intervals with five different random seeds (*x*-axis is opponent index).

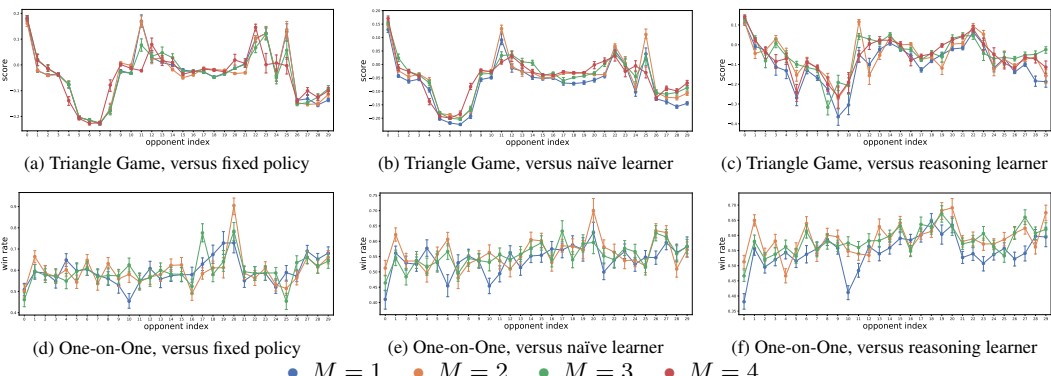

(a) Triangle Game, versus fixed policy  (b) Triangle Game, versus naïve learner  (c) Triangle Game, versus reasoning learner

(d) One-on-One, versus fixed policy  (e) One-on-One, versus naïve learner  (f) One-on-One, versus reasoning learner

● $M = 1$   ● $M = 2$   ● $M = 3$   ● $M = 4$

Figure 8: Performance against different types of opponents, *i.e.*, fixed policy, naïve learner, and reasoning learner with different $M$ (number of recursive imagination levels). The results are plotted using mean and 95% confidence intervals with five different random seeds (*x*-axis is opponent index). Higher $M$ improves the representative capability of the opponent model, but also accumulates errors. Thus, $M$ is a tradeoff between these two. $M = 1, 2, 3, 4$ are performed in Triangle Game, while $M = 1, 2, 3$ are performed in One-on-One. In general, $M \geq 2$ performs similarly, which verifies $M$ is robust. Note that $M = 1$ is MBOM w/o IOPs.

Figure 7 shows the performance of MBOM with different rollout planning horizon $k$. The selection of $k$ is a tradeoff between the environment model error and the accuracy of value estimation. In addition, the computational complexity of IOPs increases exponentially with $k$. In a sparse reward environment, appropriately increasing $k$ makes the algorithm robust. While in a dense reward environment, a smaller $k$ works well.

Figure 8 shows the performance of MBOM with different recursive imagination levels $M$. From our theoretical analysis, we know higher $M$ improves the representation capability of the opponent model, but also accumulates the model error. As illustrated in Figure 8, except $M = 1$ (*i.e.*, MBOM w/o IOPs), $M = 2, 3, 4$ perform similarly. This indicates that $M \geq 2$ is robust. Larger $M$ increases the representation capability of IOPs, but does not always improve the performance, *i.e.*, the gain is vanishing when $M$ increases due to the compounding error. Moreover, in practice $M$ also linearly increases the computational cost, thus in general smaller $M$ are preferred, *e.g.*, 2 or 3.

# D  Detailed Results on Predator-Prey

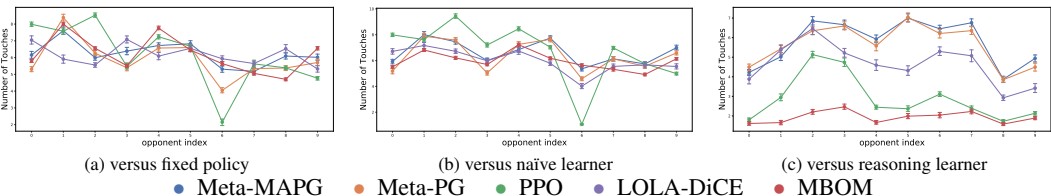

(a) versus fixed policy      (b) versus naïve learner      (c) versus reasoning learner

● Meta-MAPG    ● Meta-PG    ● PPO    ● LOLA-DiCE    ● MBOM

Figure 9: Performance against different types of opponents, *i.e.*, fixed policy, naïve learner, and reasoning learner in Predator-Prey, where *x*-axis is joint opponent index. Smaller touch number means better performance.

Figure 9 shows the performance when against different types of opponents compared with the baselines. For each type, there are ten test joint opponent policies. The results show MBOM substantially outperforms the baselines against reasoning learners. LOLA-DiCE, Meta-PG and Meta-MAPG underperform PPO when against multiple reasoning learners, indicating their adaptation may induce a negative effect on the agent performance. The reasons may be as follows. LOLA-DiCE exploits the opponent model to estimate the learning gradient of opponent policy. However, when against multiple reasoning learners, the estimated gradient of their joint policy can hardly be accurate enough to capture the change of their individual policies as each opponent learns conditioned on the agent's policy. Meta-PG and Meta-MAPG both update the agent' policy to accommodate the future policy of opponent. However, the future joint policy of multiple reasoning opponents is much harder to anticipate than in single-opponent cases.

# E  Experiment Settings

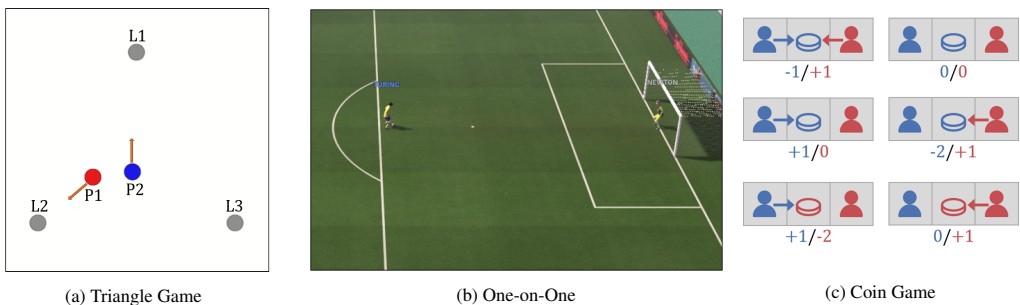

(a) Triangle Game      (b) One-on-One      (c) Coin Game

Figure 10: Illustrations of the scenarios.

The experiment environments are detailed as follows:

**Triangle Game.** As shown in Figure 10(a), there are two moving players, player 1 and 2, and three fixed landmarks, $L1 - L3$, in a square field. The landmarks are located at the three vertexes of an equilateral triangle with a side length $0.6$. When the distance between a player and a landmark is less than $0.15$, the agent touches the landmark and has the state $T$. $T1$ indicates that the player touches the landmark $L1$, and so on. If the player does not touch any landmark, the player state is $F$. The payoff matrix of the two players is shown in Table 3, where player 2 has inherent disadvantages since the optimal solution of player 2 always strictly depends on the state of player 1. When facing different policies of player 1, player 2 has to adjust its policy to adapt to player 1 for higher reward. We control player 2 as the agent and take player 1 as the opponent.

**One-on-One.** As shown in Figure 10(b), there is a goalkeeper and a shooter who controls the ball in the initial state and could dribble or shoot the ball. At the end of an episode, if the shooter shoots the ball into the goal, the shooter will get a reward $+1$, and the goalkeeper will get a reward $-1$. Otherwise, the shooter will get a reward $-1$, and the goalkeeper will get a reward $+1$. The goalkeeper could only passively react to the strategies of the shooter and makes policy adaptation when the shooter strategy changes. We control the goalkeeper as the agent and take the shooter as the opponent.

Table 3: Payoff matrix of Triangle Game.

|  |  | Player 2 | | | |
|---|---|---|---|---|---|
|  |  | F | T1 | T2 | T3 |
| Player1 | F | 0/ 0 | −0.5/ +0.5 | −0.5/ +0.5 | −0.5/ +0.5 |
|  | T1 | +0.5/ −0.5 | +1/ −1 | +1/ −1 | −1/ +1 |
|  | T2 | +0.5/ −0.5 | −1/ +1 | +1/ −1 | +1/ −1 |
|  | T3 | +0.5/ −0.5 | +1/ −1 | −1/ +1 | +1/ −1 |

Predator-Prey. We follow the setting in MPE [21]. In each game, the agent plays against three opponents (predators), and the episode length is 200 timesteps.

Coin Game. As shown in Figure 10(c), there are two players, red and blue, moving on a $3 \times 3$ grid field, and two types of coins, red and blue, randomly generated on the grid field. If the player moves to the position of the coin, the player collects the coin and receives a reward of $+1$. However, if the color of the collected coin is different from the player's color, the other player receives a reward of $-2$. The length of the game is 150 timesteps.

**Preparing opponents.** For the two types of opponents, fixed policy and naïve learner, we run independent PPO [32] algorithm for 10 times. During each run, we store 20 opponent policies in the training set, 3 opponent policies in the validation set, and 3 opponent policies in the test set. So the sizes of the training set, validation set, and test set are 200, 30, and 30, respectively. The validation set is only required by Meta-PG and Meta-MAPG. The reasoning learner learns a model to predict the action of the agent and a policy conditioned on the predicted action. Since the initial parameters of the reasoning learner should not be shared with the first two types of opponents, we train additional 30 reasoning learners in the same way aforementioned and add them to the test set.

To increase the diversity of the opponent policy, the method [37] can be adopted, but here we use some tricks to increase the diversity without incurring too much training cost. For the Triangle Game, we trained the opponent set with a modified reward, so that we could get the opponent that commuting between T1 and T2. Other types of opponents, such as hovering around a landmark, commuting between 2 landmarks, or rotating among 3 landmarks, are obtained in a similar way. For One-on-One, we set a barrier in front of the goal (invisible, but can block the ball) and only keep a gap so that the ball can enter the goal. We trained opponents with such gaps in different positions.

**Pre-training and Test.** In the pre-training phase, all methods are well trained with $\nu$ learning opponents of training set. The data during this phase is collected to fit the environment model and the level-0 IOP for MBOM. In the test phase, the agent interacts with the opponents in the test set to evaluate the ability to adapt to various opponents. The test phase lasts for 100 episodes, during which the environment model is no longer trained and the agent continuously finetunes parameters. Fixed policy, naïve learner, and reasoning learner use the test set to initialize parameters and continuously learn by respective learning ways. All methods use the same training set, validation set, and test set. There are enough opponents with different policies for testing to ensure that experimental results are unbiased.

The hyperparameters of MBOM are summarized in Table 4.

The code is available at https://github.com/PKU-RL/MBOM.

Table 4: Hyperparameters

| | | Triangle Game | One-on-One | Predator-prey | Coin Game |
|---|---|---|---|---|---|
| PPO | policy hidden units | MLP[64,32] | LSTM[64,32] | MLP[64,32] | MLP[64,32] |
| | value hidden units | MLP[64,32] | MLP[64,32] | MLP[64,32] | MLP[64,32] |
| | activation function | ReLU | ReLU | ReLU | ReLU |
| | optimizer | Adam | Adam | Adam | Adam |
| | learning rate | 0.001 | 0.001 | 0.001 | 0.001 |
| | num. of updates | 10 | 10 | 10 | 10 |
| | value discount factor | 0.99 | 0.99 | 0.99 | 0 |
| | GAE parameter | 0.99 | 0.99 | 0.99 | 0 |
| | clip parameter | 0.115 | 0.115 | 0.115 | 0.115 |
| Opponent model | hidden units | MLP[64,32] | MLP[64,32] | MLP[64,32] | MLP[64,32] |
| | learning rate | 0.001 | 0.001 | 0.001 | 0.001 |
| | batch size | 64 | 64 | 64 | 64 |
| | num. of updates | 10 | 10 | 10 | 10 |
| IOPs | num. of levels $M$ | 3 | 3 | 2 | 2 |
| | learning rate | 0.005 | 0.005 | 0.005 | 0.005 |
| | update times | 3 | 3 | 3 | 3 |
| | rollout horizon | 2 | 5 | 1 | 1 |
| | decayed factor of $\Psi$ | 0.9 | 0.9 | 0.9 | 0.9 |
| | horizon of $\Psi$ | 10 | 10 | 10 | 10 |
| | s-softmax parameter | 1 | $1.1/e$ | 1 | 1 |