# OpenReview forum: "Model-Based Opponent Modeling"
_NeurIPS.cc/2022/Conference — NeurIPS 2022 Accept_

### Official Review · Reviewer_Le9a · 2022-07-11

**Rating:** 3
**Confidence:** 4
**Soundness:** 1 poor
**Presentation:** 2 fair
**Contribution:** 2 fair

**Summary:**

This work tackles the problem of modelling agent behaviour in multiagent systems. It focuses explicitly on modelling agents that, in their decision-making process, either (i) adapt their policy to other agents' exhibited behaviour or (ii) reason about other agents' behaviour. This work proposes MBOM, a model-based recursive reasoning-based approach to model agents that reason about others' behaviour. This approach is combined with a method that mixes inferred agent policies from different levels of recursive reasoning to (a) improve the model accuracy for predicting agents' behaviour while also (b) helping model learning agents that adapt their policies to other agents' behaviour.

This work evaluates MBOM in the Triangle  Game, One-on-One, and Predator-Prey environment against two types of baseline agent modelling techniques. The first type of baselines consists of approaches designed for modelling adaptive agents that change their policies according to others' behaviour. The second baselines are ablations of MBOM designed to elucidate the importance of MBOM's main components. When a controlled agent uses MBOM and the baseline agent modelling techniques for decision making, the experiments show agents equipped with MBOM achieving higher returns than agents that use the baseline agent modelling techniques.

**Questions:**

1) Why are the baselines designed for modelling adaptive learning agents even worse than PPO when evaluated against learning agents?
2) What is the intuition why MBOM's policy mixing approach improves performance when dealing with adaptive learning agents?
3) How will MBOM perform against agents with unknown reward functions (i.e. those whose reward function highly differs from agents encountered during training)
4) How does MBOM's rollouts process cope with the exponential increase in joint action space size as the number of agents increases?
5) (Line 22) "Overall supremacy" --> Supremacy in terms of what? Maybe even be more specific on what needs to be optimised (e.g. model accuracy or returns).
6) How accurate is MBOM (and other baselines) when modelling the actions of the modelled agents?
7) How are the reasoning learner finetuned during interaction?
8) In Figure 4b), why are Meta-PG and Meta-MAPG represented by 1 line?

**Limitations:**

As mentioned in the above points, the work has not provided sufficient analysis on the limitations of MBOM to (i) deal with agents with previously unseen reward functions and (ii) to environments with more number of agents. At the same time, the work is currently fairly constricted to generic MARL setting. Thus, I do not believe it requires any additional statements on its potential societal impact.

**Strengths And Weaknesses:**

Strengths

To my knowledge, MBOM's deep model-based approach to learning an agent's optimal policy at each level of reasoning is novel. Furthermore, its approach of mixing agent policies inferred at various levels of reasoning to improve modelling accuracy is also novel. By contrast, other methods such as PR2 (Wen et al., 2019) or GR2 (Wen et al., 2019) only consider the policy resulting from the deepest level of reasoning for decision making.

The proposed ablation study over MBOM's component is designed well. The baseline selection elucidates the importance of MBOM's (i) recursive reasoning process and (ii) its usage of model mixing. In future iterations of this work, this ablation study should remain a part of the paper.

Except for the lack of descriptions regarding MBOM's model architecture in Figure 1, the model description is clear. The author's effort to explain the role of each of MBOM's components makes it easier to understand the model.


Weaknesses

This work lacks citations to older works in recursive reasoning for opponent modelling. In particular, works based on the I-POMDP framework (Gmytrasiewicz et al., 2005) should also be relevant to this work. Furthermore, an approach that applies deep neural networks to solve I-POMDPs has been explored by Han et al. (2019). In the case of the I-POMDP-Net proposed by Han et al. (2019), their approach does not rely on the CTDE learning paradigm, which the authors used to characterise prior deep learning-based works on recursive reasoning.

Major weaknesses:
1. The uncertain role of policy mixing in modeling adaptive learning agents
Despite positioning MBOM as an approach for modelling (i) adaptive learning agents and (ii) agents that also learn models of the controlled agent for decision-making, it is not clear which components of MBOM contribute toward modelling adaptive learning agents. While the authors attributed MBOM's strong performance when dealing with (i) to its policy mixing method in various parts of the manuscript (lines 195 and 279), the intuition behind why policy mixing helps in modelling (i) is not clear. More specifically, the non-parametric design of the policy mixer does not seem to help in learning the changes in modelled agent's policy resulting from the learning agent's actions.

My scepticism regarding the policy mixer's role in modelling (i) is further illustrated in the ablation study, which results are displayed in Figures 3c and 3d. Notice that there are ablations of MBOM that deliver similar performance to MBOM even without policy mixing (MBOM-{\phi_{0}}, MBOM-{\phi_{1}}, and MBOM-{\phi_{2}}) as long as the recursive reasoning process is done to an appropriate level. Comparing this to results in Figures 3a and 3b where we have an ablation of MBOM without recursive reasoning, the most significant drop in performance when dealing with adaptive learning agents results from not applying any recursive reasoning. Thus, this indicates that recursive reasoning is why MBOM models (i) well. The manuscript lacks further explanations why recursive reasoning results in improved modelling performance against (i) despite not modelling the changes in agents' policies resulting from learning.

2. Baseline selection and implementation
The baseline selection and implementation are also highly questionable. In particular, the authors selected baselines designed for modelling adaptive learning agents. Yet against adaptive learning agents, these baselines performed significantly worse than PPO and the proposed approach, which is not equipped with anything to model adaptive learning agents. This raises the question of whether these baselines are correctly implemented in the first place.

To demonstrate the need to model adaptive agents (agents that learn) as opposed to modeling just fixed agents, it would be helpful to include agent modeling baselines which were not specifically designed to model learning agents, such as LIAM:
Georgios Papoudakis, Filippos Christianos, Stefano V. Albrecht. Agent Modelling under Partial Observability for Deep Reinforcement Learning. NeurIPS 2021

Although previous works model (i) with recursive reasoning, there is a lack of recursive reasoning baselines. While the authors mentioned other recursive reasoning methods' reliance on the CTDE paradigm as justification for not comparing against them (line 89), I highly believe a comparison against these methods must be made. Since the learning agent decides its action without any centralised component during execution, we can see this as utilising privileged information that only exists during training. Even MBOM uses some privileged information that is not ordinarily accessible during execution (e.g. modelled agent's rewards) for training. In the worst case, recursive reasoning methods not based on CTDE training like I-POMDP-Net can be used as a baseline.

3. Robustness to diverse opponents
While one of the central claims of this work is that MBOM allows the learning agent to perform well against a diverse set of opponents, MBOM has not been evaluated against agents whose reward function is highly different to those encountered during training. This is particularly important since many applications require learning agents to deal with previously unseen decision-makers whose reward functions are unknown (e.g. humans in autonomous driving scenarios). In this case, an experiment to evaluate MBOM's capability in those scenarios is also necessary.

4. Lack of experiments against agents with deeper levels of recursive reasoning.
While the reasoning learner is an example of agents that model the controlled agent, there is a lack of evaluation against agents with deeper levels of recursive reasoning. A potential improvement is to evaluate MBOM against another MBOM agent with various depths of reasoning.

5. A lack of analysis on model accuracy
The current work's analysis only reports the returns resulting from applying MBOM and the baselines. Nevertheless, it is important to report the model accuracy in any work in opponent modelling. Future iterations of this work can measure the log-likelihood of modelled agents' predicted actions.

6. Scalability to scenarios with more opponents.
It would be interesting to see whether MBOM scales to scenarios with more modelled agents. In particular, the rollout procedure done at every level of recursive reasoning requires more rollouts as the joint action space increases. Yet, the experiments have been limited to scenarios with small number of agents.

Clarity
1. Imprecise statements.
Overall, there are a few statements that are imprecise in the manuscript. Since these may potentially confuse readers, I recommend fixing these highlighted sentences:

(Line 8) "All kinds of opponents" --> Be precise on the type of agents that are evaluated in this work's experiments.
(Line 20-21) "interacting with diverse opponents makes the environment nonstationary from the agent's perspective." --> This is only true when other agents' policies are changing. When agents' policies are fixed, one can account for the effect of agents' actions in the transition function. Even when agents' policies are unknown, the problem can be seen as a POMDP as long as these policies remain fixed.

2. Lack of details in Figure 1
If done correctly, Figure 1 can help readers understand the proposed approach. However, the lack of captions in Figure 1 explaining the components of MBOM makes it challenging to understand the Figure. Also, consider adding labels associated to the dashed boxes to indicate which components of MBOM they represent.

3. Justification on baseline and opponent design
The experiment section can be improved by highlighting why specific baselines or opponents are designed the way they are. Focusing on the insights gained from comparisons against specific baselines or experiments using certain opponents can also help highlight the claims provided in this work.


Significance

While this work presents an interesting approach that has potentially major significance for people working in MARL and opponent modelling, further experiments and comparisons are required to fully demonstrate its use in modelling (i) adaptive learning agents and (ii) agents that also learn models of the learning agent. Its limited comparison against prior recursive reasoning methods and the lack of analysis in terms of model accuracy particularly stands out as why this work has limited significance as is.

---

> ### Author Response · Authors · 2022-08-02
> **Response to Reviewer Le9a (Part Ⅰ)**
>
> Thanks for your valuable comments. As follows, we address your concerns in detail.
>
> > Why are the baselines designed for modeling adaptive learning agents even worse than PPO when evaluated against learning agents?
>
> **First it is worth noting that PPO is significantly inferior to Meta-MAPG and Meta-PG (Table 1) in Triangle Game and One-on-one.**
>
> MBOM intends to adapt learning agents in the interaction (execution) phase. So we chose appropriate environmental settings and state-of-the-art baselines in this domain. There are only 100 episodes of the interaction phase. Its opponent comes from the test set, which is different from the training set in pre-training. We compare Meta-MAPG, Meta-PG, LOLA-DiCE, and PPO as baselines. Meta-MAPG and Meta-PG are state-of-the-art methods in the domain of adaptive learning agents in the interaction phase, and those performances are generally better than PPO.
>
> In Predator-Prey, Meta-MAPG and Meta-PG do not adapt to the changes of opponents’ policies only when against reasoning learners. Since the adaptation ability of Meta-MAPG and Meta-PG is highly related to the policies of opponents in the pre-training phase. In the experiments, the opponents in the pre-training phase are not reasoning, which will much differ from the policies of reasoning learners in the interaction phase. Specifically, all the opponents in the training phase are strong opponents, while in the interaction phase the reasoning learners would make wrong reasonings and become mediocre. The difference between policies in the pre-training phase and the interaction phase destroys the adaptation ability of Meta-MAPG and Meta-PG.
>
> LOLA-DiCE is the classical method to consider the learning of the opponent during the pre-training phase. It cannot adapt to the opponent during the interaction. It is reasonable that LOLA-DiCE shows worse performance than PPO in the interaction phase.
>
>
> > What is the intuition why MBOM's policy mixing approach improves performance when dealing with adaptive learning agents?
>
> For the mixer, essentially, $\alpha$ is the weight of mapping IOPs to the true policy of the opponent. Since the opponent is adaptive learning, and the learning process is unknown , a single IOP might erroneously estimate the opponent. To obtain stronger representation capability and accurately capture the learning opponent, we linearly combine the IOPs to get a mixed IOP. That is the reason why policy mixing approach improves performance when dealing with adaptive learning agents.
>
> Recursive Imagination is different from the classical Recursive Reasoning. In Recursive  Imagination, the agent simulates the interactions with opponents in the environment model and infers the opponent's policy. The inferred opponent policy is called Imagined Opponents Policy (IOP). This simulation happens in the environment model, so we call it "imagination".  **It is unattainable that the recursive reasoning process is done to an appropriate level, because the agent does not know which IOP is the appropriate one.** Yet the Bayesian Mixing maps IOPs to be close to the true policy of the opponent by adjusting $\alpha$ using the observed opponent's actions in interaction.
>
> In the ablation study, we verify that the mixer is helpful for performance improvement and stability. Performance improvement is reflected in Figure 3c and 3d, where MBOM outperforms MBOM without policy mixing (MBOM-${\\phi\_{0}}$, MBOM-${\\phi\_{1}}$, and MBOM-${\\phi\_{2}}$). Performance stability is reflected in Figure 3c and 3d, where MBOM performs robustly on all types of opponents (fixed policy, naive learner, reasoning learner). While MBOM targets learning opponents (naive learner and reasoning learner), it still has no defect for fixed policy opponents. In Figure 3c, against reasoning learner, MBOM-${\\phi\_{2}}$ performs lower than MBOM-${\\phi\_{1}}$, which shows that merely increasing the level of reasoning does not always work. The mixer can avoid the negative effects of increasing the level of reasoning.

---

> ### Author Response · Authors · 2022-08-02
> **Response to Reviewer Le9a (Part Ⅱ)**
>
> > How will MBOM perform against agents with unknown reward functions (i.e. those whose reward function highly differs from agents encountered during training)
>
> This is out of scope. As described in Section 3 (lines 120-130), the problem is defined as $n$-agent stochastic game, where the MDP is fixed. Moreover, MBOM is a novel method for "adapting to changing opponent policies". The loose constraints on the opponent's learning make it possible to apply to general-sum game, as shown in Eq.4, but it may not work always. In order to focus on "adapting to changing opponent policies", we cast the problem to zero-sum games and cooperative games, as others work, like [11,12,13].
>
> [11] Ying Wen, Yaodong Yang, Rui Luo, Jun Wang, and Wei Pan. Probabilistic Recursive Reasoning for Multi-Agent Reinforcement Learning. ICLR, 2019.
>
> [12] Jakob Foerster, Richard Y Chen, Maruan Al-Shedivat, Shimon Whiteson, Pieter Abbeel, and Igor Mordatch. Learning with Opponent-Learning Awareness. AAMAS, 2018.
>
> [13] Kaiqing Zhang, Sham M. Kakade, Tamer Basar, and Lin F. Yang. Model-based multi-agent RL in zero-sum markov games with near-optimal sample complexity. NeurIPS 2020.
>
> > How does MBOM's rollouts process cope with the exponential increase in joint action space size as the number of agents increases?
>
> The results in Predator-Prey show that it is feasible for MBOM to treat all opponents as a joint opponent. The consequent problem is that the computational cost increases exponentially with the number of opponents. One potential solution is parallel computing. Computationally efficient implementations are also an open problem for the agent modeling community. It is a limitation of MBOM that we have discussed in Lines 652-655.
>
> > (Line 22) "Overall supremacy" --> Supremacy in terms of what? Maybe even be more specific on what needs to be optimised (e.g. model accuracy or returns).
>
> MBOM intends to achieve a higher return by improving model accuracy via adapting to changing opponents. The results of comparison with MBOM-pro (Common Question 2) illustrate that an accurate opponent model can indeed improve returns, and the results of Common question 1 reflect that MBOM has higher model accuracy than MBOM w/o IOPs.
>
> > How accurate is MBOM (and other baselines) when modelling the actions of the modelled agents?
>
> **Please refer to the response to common questions 1.**
>
> > How are the reasoning learner finetuned during interaction?
>
> The reasoning learner (opponent) learns a model to predict the action of the agent and a policy net conditioned on the predicted action of the agent, both of which are updated during the interaction. The model is updated by supervised learning using the observed agent's action, and the policy is updated using PPO.
>
> > In Figure 4b), why are Meta-PG and Meta-MAPG represented by 1 line?
>
> The Cooperative Task in our experiments follows the setting of LOLA, where both two agents learn using MBOM (or other baselines) from the scratch to adapt to each other without pre-training. Meta-PG and Meta-MAPG degenerate to Policy Gradients in this task as there is no training set (Lines 326-327). So we use 1 line to show their common results.
>
> >  Lack of modeling and recursive reasoning baselines.
>
> **Please refer to the response to common questions 2.**
>
> > Lack of experiments against agents with deeper levels of recursive reasoning.
>
> We added the experiment using a deeper reasoning learner as the opponent in Triangle Game. The results are shown in Table 3. Additionally, human beings show a limited amount of working memory (1 - 2 levels on average) in strategic thinking [14], so it is reasonable to usually select smaller M.
>
> Table 3. Results of deeper reasoning opponent.
>
> |     | Fixed policy  | Naive learner  | Reasoning learner (M=1)  | Deeper reasoning learner (M=2) |
> |  :----  | :----:  | :----:  | :----:  | :----:  |
> |Meta-MAPG |	-2.01 (0.06) |	-5.76 (0.29) |	-6.14 (0.84)| 	**-14.88 (5.82)**|
> |Meta-PG |	-3.78 (0.13) |	-6.72 (0.18) |	-8.35 (1.87)| **-12.23 (4.0)**|
> |LOLA-DiCE |	-22.51 (5.22) |	-20.48 (4.02) |	-21.55 (4.43)| **-25.43 (4.43)**|
> |MBOM |	-1.19 (0.03) |	-1.66 (0.29) |	-2.75 (0.89)| 	**-1.83 (0.55)**|
>
> [14] Giovanna Devetag and Mas- simo Warglien. Games and phone numbers: Do short-term memory bounds affect strategic behavior, Journal of Economic Psychology, 24(2):189–202, 2003.

---

### Official Review · Reviewer_tVUV · 2022-07-11

**Rating:** 6
**Confidence:** 4
**Soundness:** 3 good
**Presentation:** 3 good
**Contribution:** 3 good

**Summary:**

This paper proposes a model-based opponent modeling method to handle the interaction with other agents in multiagent systems. The main contribution of the proposed method is that it can adapt to different kinds of opponent policy, e.g., fixed-type, adaptive-type, and reasoning-type. Concretely, the proposed method first lets the agent interact with diverse opponents and collects the interaction experience. Then it uses the collected experience to train an environment model. Using this environment model, the agent imagines the adaption of the opponent policy and uses the imagined best response of the opponent to refine the opponent model. The agent repeats this process to get a set of opponent models reflecting different recursive reasoning levels. The authors further use Baysian mixing to get the final opponent model based on the opponent model set. As the Bayesian mixing is non-parametric, the final opponent model can quickly adapt to the true dynamic opponent policy.

**Questions:**

1. In the proposed MBOM, the authors pre-train the agent’s policy using PPO while interacting with $v$ different opponents with diverse policies. However, I am surprised that the PPO can train a good agent policy in this way. In my opinion, the RL training highly relies on the sampled experience. The training process is trying to find a (local) optimal policy that can maximize the expected return given the sampled experience. Therefore, when trained with a certain type of opponent A, it should be easy for PPO to learn a good policy. However, if a different type of opponent B joins the training process, it will affect the interaction experience, i.e., in some episodes, the agent interacts with opponent A while in other episodes the agent interacts with opponent B. In this case, the PPO will adjust the policy trained with opponent A because it needs to consider the experience corresponding to opponent B. As a consequence, the new policy will likely have worse performance when facing opponent A especially when opponent A is very different from opponent B. Therefore, I would like to know more details about how to avoid this kind of issue and learn a good policy when interacting with $v$ different opponents with diverse policies. For example, what is the input of agent policy in every step during the training? I think the agent policy pre-training is critical because, during the recursive imagination, the agent policy is fixed and plays an important role.

2. Lemma 2 is critical for the theoretical analysis. However, is it a practical assumption? In what kind of scenarios this assumption will hold?

3. The authors use LOLA-DiCE as one baseline because it considers the potential learning of opponents. However, if the assumptions required by LOLA-DiCE do not hold, the consideration of opponent learning may bring adverse effects instead of benefits. This is also observed in Figure 2 where LOLA-DiCE has almost the worst performance in every case. Therefore, it may be better if the authors can also compare with other opponent modeling works, e.g. [27], [30]. Although the authors think they only consider opponent policies that are independent of the agent, sometimes simpler settings can be better than complex settings like in LOLA-DiCE.

4. MBOM has a pre-train phase where it sees a set of opponents with diverse policies. In comparison, Meta-PG only uses trajectories from the current opponents. Do the pre-train phase brings more information to MBOM and thus, gives it some advantages when interacting with the current opponents?

**Limitations:**

1. The assumption in Lemma 3 seems strong. Although the authors claim that "larger M improves the representation capability of IOPs and thus better satisfies the assumption in Lemma 3", larger M also brings higher approximation error (which hinders the representation capability). Therefore, there is no guarantee that the assumption in lemma 3 can hold. If it does not hold, the Bayesian mixing may not approximate the true probability of the opponent well. Moreover, the recursive imagination will incur a considerable amount of computational cost.

2. There are duplicate texts on page 6: "However, larger M also has advantages. To analyze, we first define the benefit using the mixed IOP as..."

**Strengths And Weaknesses:**

Strengths: this paper tackles a challenging opponent modeling problem and the proposed method is a good combination of well-known techniques. The existing works are discussed in detail and this paper situates itself in the literature well. The presentation of this paper is also clear. The experiment results are sound and support the authors' claim.

Weaknesses: some technical parts need more description and explanation. Some existing works may also be worthy to compare.

---

> ### Author Response · Authors · 2022-08-02
> **Response to Reviewer tVUV**
>
> Thank you for acknowledging our novel contributions as well as raising valuable questions.
>
> > How to avoid the proposed issue and learn a good policy when interacting with different opponents with diverse policies. For example, what is the input of agent policy in every step during the training?
>
> In pre-training, the agent is only to learn a good policy. The input to the agent's policy net is $(s, a\^o)$, where $a\^o$ is the real action distribution of the opponent at the current timestep. Under such conditions, the agent's policy learning is a single-agent MDP problem, and the agent's experience buffers are from multiple opponents. Unlike your case, it is rare for oscillations to occur under such conditions that the agent can access the opponent's policy.
>
> > Lemma 2 is critical for the theoretical analysis. However, is it a practical assumption? In what kind of scenarios this assumption will hold?
>
> Lipschitz continuous is a common and practical assumption in reinforcement learning. More specifically, Lipschitz continuous is defined as $\|\|f(x\_2)-f(x\_1)|| \\leq K\|\|x\_2-x\_2\|\|$ and can be interpreted as the derivatives of function $f$ between two variables are controlled to not exceed a constant $K$. The geometric meaning is that function $f$ does not change suddenly and excessively. In lemma 2, we assume that the value function has Lipschitz continuous, which means that the values of the neighboring states do not differ too much and their differences do not exceed a constant K. Intuitively, this assumption will hold in various stable environments, and many existing works are also based on this assumption, e.g., [8,9,10]. We believe this assumption is sufficiently practical.
>
> [8] Asadi K, Misra D, Littman M. Lipschitz continuity in model-based reinforcement learning. ICML 2018.
>
> [9]Luo Y, Xu H, Li Y, et al. Algorithmic framework for model-based deep reinforcement learning with theoretical guarantees. ICLR 2019.
>
> [10]Curi S, Berkenkamp F, Krause A. Efficient model-based reinforcement learning through optimistic policy search and planning. NeurIPS 2020.
>
> > The authors use LOLA-DiCE as one baseline because it considers the potential learning of opponents. However, if the assumptions required by LOLA-DiCE do not hold, the consideration of opponent learning may bring adverse effects instead of benefits. This is also observed in Figure 2 where LOLA-DiCE has almost the worst performance in every case. Therefore, it may be better if the authors can also compare with other opponent modeling works, e.g. [27], [30]. Although the authors think they only consider opponent policies that are independent of the agent, sometimes simpler settings can be better than complex settings like in LOLA-DiCE.
>
> **Please refer to the response to common questions 2.**
>
> > MBOM has a pre-train phase where it sees a set of opponents with diverse policies. In comparison, Meta-PG only uses trajectories from the current opponents. Do the pre-train phase brings more information to MBOM and thus, gives it some advantages when interacting with the current opponents?
>
> All algorithms (MBOM and baselines) use the same training set and test set. Thus, Meta-PG has also seen a set of opponents with diverse policies in the pre-training phase. Therefore, it is a **fair** comparison.
>
> > The assumption in Lemma 3 seems strong. If it does not hold, the Bayesian mixing may not approximate the true probability of the opponent well. Moreover, the recursive imagination will incur a considerable amount of computational cost.
>
> According to Theorem 1, a larger M does imply a larger cumulative error, but as we argued in line 235-239, the choice of M is a tradeoff between representation capability and the error. We insist that an appropriate increase in M can bring more benefits and thus compensate the influence of the error. Intuitively, if the opponent's level is not in our given set of IOPs, it is difficult to fit the opponent's real policy no matter how we do the reasoning. A proper increase of M can better ensure that our set of IOPs contains the opponent's level, thus ensuring the effectiveness of reasoning. We also demonstrate this experimentally. As shown in Figure 8 and line 583-589 of Appendix C, we did an ablation study on the parameter M. The conclusion shows that the best results can be achieved when M takes the value of 2 or 3, where the computational cost is not high.

---

> > ### Comment · Reviewer_tVUV · 2022-08-06
> > **Thanks for the response**
> >
> > The authors have cleared my concerns with the detailed response.

---

### Official Review · Reviewer_yT2T · 2022-07-14

**Rating:** 7
**Confidence:** 4
**Soundness:** 3 good
**Presentation:** 3 good
**Contribution:** 3 good

**Summary:**

This paper introduces an approach model-based opponent modeling (MBOM) for deep multiagent reinforcement learning (MARL) that combines (1) recursive imagination to estimate the reasoning of other agents at different hierarchical levels of reasoning (common in Theory of Mind reasoning like planning-based I-POMDPs, but novel in RL) and (2) Bayesian mixing that estimates the collective behaviors of all other agents in the system as a mix of the lower level recursive reasoning.  Such an approach enables the agent to learn how to behave when its oppponents follow static or adaptive policies without requiring explicit models of the neighbor's reasoning (including learning algorithm or learning gradients).  Theoretical analysis establishes error bounds on the value estimates.  Empirical evaluation on zero-sum games, competitive, and cooperative tasks from MPE demonstrate the advantages of the approach over appropriate baselines.


**Questions:**

1) I didn't quite follow how \alpha_m = p(m | a^o).  Shouldn't \alpha_m (the mixing proportion) simply be p(m)?  Otherwise, shouldn't Eq. 6 also have a sum over all possible actions (if the policy is stochastic)?

**Limitations:**

The limitations of the approach are adequately described.

**Strengths And Weaknesses:**

Overall, the research will be of interest to the popular MARL community at NeurIPS.  The approach is novel and well evaluated.  The paper is relatively easy to follow and has a thorough literature review.  I appreciated the inclusion of both theoretical and empirical analysis to evaluate the approach and derive key properties.  The supplement provided important information for reproducibility.

---

> ### Author Response · Authors · 2022-08-02
> **Response to Reviewer yT2T**
>
> Thank you for acknowledging the novelty and contributions.
>
> > I didn't quite follow how \alpha_m = p(m | a^o). Shouldn't \alpha_m (the mixing proportion) simply be p(m)? Otherwise, shouldn't Eq. 6 also have a sum over all possible actions (if the policy is stochastic)?
>
> The policy is stochastic. However, only one executed action $a\^o\_t$ can be observed by the agent after timestep $t$ during the interaction phase, which makes the observed opponent's policy deterministic.
>
> In such case, $\\alpha \\propto p(m)$ is equivalent to $\\alpha \\propto p(m\|a\^o)$.
>
> $$\begin\{aligned\}
>     \\alpha \\propto p(m) \\implies \\Psi\_m\^t \&= \\sum\_{l=t-H}\^{t-1}\\lambda\^{t-l}p(m) \\\\
>     \&= \\sum\_{l=t-H}\^{t-1}\\sum\_{a^o}\\lambda\^{t-l}p(m\|a\_l\^{o})p(a\_l\^o) \\\\
>     \&= \\sum\_{l=t-H}\^{t-1}\\lambda\^{t-l}p(m\|a\_l\^{o}) \\implies \\alpha \\propto p(m\|a\^o)
> \end{aligned}$$

---

> > ### Comment · Reviewer_yT2T · 2022-08-08
> > **RE: Author's Responses**
> >
> > I thank the authors for their responses to our questions and comments.

---

### Official Review · Reviewer_PoB7 · 2022-07-15

**Rating:** 4
**Confidence:** 4
**Soundness:** 2 fair
**Presentation:** 3 good
**Contribution:** 2 fair

**Summary:**

This paper proposes recursive reasoning to model the opponents in a multi-agent environment, especially when the opponents are capable to learn and reason. The proposed method MBOM models the environment and the joint opponents. Particularly, MBOM simulates the recursive reasoning process and fine-tune the opponent models on multiple levels. The multiple-level models are combined by a Bayesian mixing strategy. The experiments are performed extensively and show improved performance on several benchmarks.

**Questions:**

1. Line 288-290 “MBOM w/o IOPs obtains similar results to MBOM when facing fixed policy opponents because /phi_0 could accurately predict the opponent behaviors if the opponent is fixed.” and similar claims that the model can accurately predict sth. It will be more convincing if the ablation study is directly performed on the prediction accuracy of the opponent model, e.g. using a controlled opponent, since the final performance of RL agents can be affected by many factors other than the opponent model accuracy;

2. Regarding the Theorem 2, how do you define the true probability of the opponent since the opponent's policy is changing based on its own reasoning? If correct, the proof of Theorem 2 can be also generalized to the regular reasoning learners, since given sufficient data, the regular model can also approximate the true distribution;

3. Line 556-576, the ablation study regarding the weights /alpha learned during the training is a bit confusing. In the task of Triangle Game, the scores of the task facing fixed policy opponents are the highest compared to the other two tasks facing naive learner and reasoning learner, however, in the ablation study it claims that (line 563-564) the models of level-0 IOP or level-1 IOP are not accurate;

4. Is it possible that MBOM has better performance because it uses a combination of several models. It would be more convincing if the baselines involve a corresponding ensemble of models;

5. As mentioned in the Future work, “MBOM implicitly assumes that the relationship between opponents is fully cooperative”, which is very tight and significantly limits the application of MBOM;


**Limitations:**

It would be more convincing if the ablation study on the model accuracy is directly performed. In addition, personally, I would like to see a baseline with the same number of ensemble models.

**Strengths And Weaknesses:**

Strengths:
1. The paper aims to solve an interesting and important problem that how to model the reasoning and learning opponents in multi-agent tasks. The study can inspire the community potentially;
2. The paper is well presented and the experiments are extensive;

Weakness:
1. The proof of MBOM is not convincing and may present some problematic deductions;
2. The experiments cannot fully prove the effectiveness of the proposed method, especially it is hard to claim the MBOM has learned a more accurate model of the opponents even though the improved performance. Some more specific ablation study needs to be performed.

---

> ### Author Response · Authors · 2022-08-02
> **Response to Reviewer PoB7**
>
> Thanks for your valuable comments. As follows, we address your concerns in detail.
>
> > Line 288-290 “MBOM w/o IOPs obtains similar results to MBOM when facing fixed policy opponents because /phi_0 could accurately predict the opponent behaviors if the opponent is fixed.” and similar claims that the model can accurately predict sth. It will be more convincing if the ablation study is directly performed on the prediction accuracy of the opponent model, e.g. using a controlled opponent, since the final performance of RL agents can be affected by many factors other than the opponent model accuracy;
>
> **Please refer to the response to common questions.**
>
> > Regarding the Theorem 2, how do you define the true probability of the opponent since the opponent's policy is changing based on its own reasoning? If correct, the proof of Theorem 2 can be also generalized to the regular reasoning learners, since given sufficient data, the regular model can also approximate the true distribution;
>
> As we described in Eq. (7) and lines 184-187, p(m|a) is the estimated probability of **the opponent's reasonning level**. The implication of Theorem 2 is that when given enough data, we can more accurately estimate the opponent's true level rather than the opponent's true policy. Although the opponent's policy may change dynamically, the distribution of opponent's level may not change. Therefore, no matter what the opponent's policy is, we can approximately estimate the distribution of opponent's level, which is the conclusion of Theorem 2.
>
> > Line 556-576, the ablation study regarding the weights /alpha learned during the training is a bit confusing. In the task of Triangle Game, the scores of the task facing fixed policy opponents are the highest compared to the other two tasks facing naive learner and reasoning learner, however, in the ablation study it claims that (line 563-564) the models of level-0 IOP or level-1 IOP are not accurate;
>
> First, it is worth noting that alpha learning is performed in the **interaction** phase, not in the pre-training phase. As IOPs are learned in the interaction phase, where the opponent is continuously updating (naive learner and reasoning learner), so it is difficult to continuously capture the opponent's policy with a small number of samples. That is the reason why the models of level-0 IOP or level-1 IOP are not accurate. The mixed IOP with alpha is much closer to the real opponent's opponent, which is the key to get high scores. This was also verified by the results testing the accuracy of the opponent model (Common question 1).
>
> > Is it possible that MBOM has better performance because it uses a combination of several models? It would be more convincing if the baselines involve a corresponding ensemble of models.
>
> **We indeed have compared two baselines, MBOM-BM and MBOM-unif, that use an ensemble of several models in the ablation study.** MBOM-BM, in Figures 3(a) and 3(b), uses random actions to finetune the IOPs rather than using the best response. MBOM-unif, in Figures 3(c) and 3(d), uniformly mixes the IOPs. MBOM outperforms the two baselines.
>
> > As mentioned in the Future work, “MBOM implicitly assumes that the relationship between opponents is fully cooperative”, which is very tight and significantly limits the application of MBOM;
>
> Multi-player games with cooperative-competitive relationship are still an open problem in the community. MBOM, like existing opponent modeling approaches, e.g., [3,4,5,6,7], cannot address such problems. How to effectively deal such problems is truely out of scope and left as future work.
>
> [3] He He, Jordan Boyd-Graber, Kevin Kwok, and Hal Daumé III. Opponent Modeling in DeepReinforcement Learning. ICML, 2016.
>
> [4] Zhang-Wei Hong, Shih-Yang Su, Tzu-Yun Shann, Yi-Hsiang Chang, and Chun-Yi Lee. A Deep Policy Inference Q-Network for Multi-Agent Systems. AAMAS, 2018.
>
> [5] Roberta Raileanu, Emily Denton, Arthur Szlam, and Rob Fergus. Modeling others using oneself in multi-agent reinforcement learning. ICML, 2018.
>
> [6] Maruan Al-Shedivat, Trapit Bansal, Yuri Burda, Ilya Sutskever, Igor Mordatch, and Pieter Abbeel. Continuous Adaptation Via Meta-Learning in Nonstationary and Competitive Environments. ICLR, 2018.
>
> [7] Dong-Ki Kim, Miao Liu, Matthew Riemer, Chuangchuang Sun, Marwa Abdulhai, Golnaz Habibi, Sebastian Lopez-Cot, Gerald Tesauro, and Jonathan P. How. A Policy Gradient Algorithm for Learning To Learn in Multiagent Reinforcement Learning. ICML, 2021.

---

### Author Response · Authors · 2022-08-02
**Common Questions**

> Ablation study for proving that MBOM can learn a more accurate model of the opponents.

We tested the prediction accuracy of opponent model in the Triangle Game, using KL divergence to measure the distance between the predicted action distribution of the opponent model and the real action distribution of opponent. The results compared with MBOM w/o IOPs (that only uses $\phi\_0$ as the opponent model without recursive imagination) are shown in Table 1 (mean and std over 5 runs). Overall, MBOM's opponent model achieves a more accurate prediction of the opponent policy than MBOM w/o IOPs.

Moreover, we tested MBOM-pro, which uses the true actions of the opponent as the input of policy during the interaction phase instead of the actions predicted by IOPs. MBOM-$pro$ can be seen as the upper bound of MBOM. It also shows that a more accurate opponent model could improve performance, as shown in Table 2.

Table 1. Prediction accuracy of opponent model.

|     | Fixed policy  | Naive learner  | Reasoning learner  |
|  :----  | :----:  | :----:  | :----:  |
| MBOM w/o IOPs  | 15.72 (0.18) | 15.17 (0.17) | 8.19 (0.09) |
| MBOM  | 15.47 (0.04) | 13.47(0.43) | 7.81(0.38) |

> Including agent modeling baselines which were not specifically designed to model learning agents.

We add a new baseline, LIAM [1], which learns representations of opponent policy by reconstructing the state-action trajectories of the other agent with its state-action trajectories to assist policy during the pre-training phase. VAEOM [2] is similar to LIAM. LIAM is not a method of adaptation to the learning opponent. When the agent learns against the opponents with different policies encountered during training, the reconstruction module of LIAM does not work well, leading to low performance, as shown in Table 2. The major reason probably is that reconstructing module did not learn the representations between different opponent actions, which can be evidenced by that the loss of the reconstructing module dropped sharply and then was always near 0.

Other recursive-reasoning-based methods are mostly not designed for adaptive learning agents during interaction. Take I-POMDP-net as an example, which uses a sampling-based representation of interactive belief state. For example, if we use $N$ samples to represent a 0-level belief, e.g. a distribution over true state (or transition model, observation model, and reward function of other agents, etc.), then each sample is a state. Similarly, we continue to use $N$ samples to represent a 1-level belief which is a distribution over 0-level belief. However, each 0-level belief consists of $N$ state samples. So the size of the representation of a 1-level belief will be $O(N\^2)$. When extended to the $l$-level I-POMDP-net, the size will be $O(N\^l)$. (The belief described in this sample is not exactly the same as in the I-POMDP-net paper, but helps us calculate complexity more intuitively.) In high-dimension continuous environment, we may need more samples $N$ to represent a distribution and the size grows exponentially with the level of reasoning. The extremely high cost of the pre-training phase makes it difficult to apply in adapting to the opponent.

[1] LIAM: Georgios Papoudakis, Filippos Christianos, Stefano V. Albrecht. Agent Modelling under Partial Observability for Deep Reinforcement Learning. NeurIPS 2021

[2] Georgios Papoudakis and Stefano V Albrecht. Variational Autoencoders for Opponent Modeling in Multi-Agent Systems. AAAI-20 Workshop.

Table 2. Performance on Triangle Game.

|     | Fixed policy  | Naive learner  | Reasoning learner  |
|  :----  | :----:  | :----:  | :----:  |
|Meta-MAPG |	-2.01 (0.06) |	-5.76 (0.29) |	-6.14 (0.84)|
|Meta-PG |	-3.78 (0.13) |	-6.72 (0.18) |	-8.35 (1.87)|
|LOLA-DiCE |	-22.51 (5.22) |	-20.48 (4.02) |	-21.55 (4.43)|
|LIAM |	$-49.61 (0.13)$ |	$-47.96 (0.6)$ |	$-49.4 (0.18)$ |
|PPO |	-13.29 (1.8) |	-18.42 (1.28) |	-20.51 (1.8) |
|MBOM |	-1.19 (0.03) |	-1.66 (0.29) |	-2.75 (0.89)|
|MBOM-$pro$ |	$0.96 (0.03)$ |	$-0.11 (0.18)$ |	$0.82 (0.35)$|

---

### Author Response · Authors · 2022-08-08
**Looking forward to feedback**

Dear Reviewers,

We first would like to thank the reviewers' efforts and time in reviewing our work. We were wondering if our responses have resolved your concerns as the author-reviewer discussion is ending soon. We will be happy to have further discussions with the reviewer if there are still some remaining questions!

Best regards,

The authors

---

### Meta-Review · Area_Chair_7yyj · 2022-08-26

**Recommendation:** Accept
**Confidence:** Less certain

**Metareview:**

This paper tackles the problem of modeling agents that are simultaneously learning or are able to reason during interaction. The proposed approach employs an environment model to simulate the opponent's reasoning process. The initial reviews are split and main concern is that the results do not adequately demonstrate that the proposed approach actually models opponents better. I believe the added ablation study and baselines have adequately addressed the concerns. Two reviewers also support acceptance after discussion (the other two didn't respond). I believe the work tackles and important problem in MARL and would spur valuable discussion at the conference. Thus, I'm leaning towards acceptance.

**Award:**

No

---

### Decision · Program_Chairs · 2022-09-14

Accept